# Experimentally determined traits shape bacterial community composition one and five years following wildfire

Dana B. Johnson [1], Jamie Woolet[1,2], Kara M. Yedinak[3] & Thea Whitman [1] ✉

Wildfires represent major ecological disturbances, burning 2–3% of Earth's terrestrial area each year with sometimes drastic effects above- and belowground. Soil bacteria offer an ideal, yet understudied system within which to explore fundamental principles of fire ecology. To understand how wildfires restructure soil bacterial communities and alter their functioning, we sought to translate aboveground fire ecology to belowground systems by determining which microbial traits are important post-fire and whether changes in bacterial communities affect carbon cycling. We employed an uncommon approach to assigning bacterial traits, by first running three laboratory experiments to directly determine which microbes survive fires, grow quickly post-fire and/or thrive in the post-fire environment, while tracking $CO_2$ emissions. We then quantified the abundance of taxa assigned to each trait in a large field dataset of soils one and five years after wildfires in the boreal forest of northern Canada. We found that fast-growing bacteria rapidly dominate post-fire soils but return to pre-burn relative abundances by five years post-fire. Although both fire survival and affinity for the post-fire environment were statistically significant predictors of post-fire community composition, neither are particularly influential. Our results from the incubation trials indicate that soil carbon fluxes post-wildfire are not likely limited by microbial communities, suggesting strong functional resilience. From these findings, we offer a traits-based framework of bacterial responses to wildfire.

Wildfires are a major ecosystem disturbance, burning 2–3% of Earth's terrestrial area annually[1], including millions of hectares in North American boreal forests every year[2]. In this region, the severity of wildfires and area burned are both increasing, driven by climate change[3,4]. During wildfires, belowground soil temperatures can exceed 500 °C, driving carbon (C) and nitrogen (N) loss via volatilization of organic matter (OM)[5] and causing a restructuring of soil bacterial communities[6]. Because soil bacteria mediate biogeochemical cycling, the role of fire in restructuring soil bacterial communities is an important component

of understanding nutrient cycling and C storage in boreal forests. To understand the effects of changing wildfire regimes on soil bacteria, traits-based frameworks offer the potential utility of integrating bacteria into biogeochemical models using traits representing the extremely high diversity of soil bacteria within a few manageable parameters[7].

While fire ecology is well-developed for aboveground organisms[8], fire ecology frameworks for soil microbes trail far behind (despite certain well-known fire-responsive fungi, such as morels[9]). Grime's strategic framework for plants[10] has shaped understanding of plant

[1]University of Wisconsin-Madison, Madison, WI, USA. [2]Colorado State University, Fort Collins, CO, USA. [3]Forest Products Laboratory, USDA Forest Service, Madison, WI, USA. ✉e-mail: twhitman@wisc.edu

recovery following disturbance. This framework suggests that plants can be classified according to the relative importance of three primary strategies: competitive plants, stress-tolerant plants and ruderal plants. There is growing interest in extending these ideas to microbial ecology. Building on Grime's framework, Malik et al. proposed a trait-based framework for soil microbial ecology with three microbial C acquisition life strategies: high yield, resource acquisition and stress tolerance[11]. A modified version of Grime's framework may also be helpful in untangling the role of fire in structuring soil microbial communities in ecosystems adapted to and dependent upon wildfire. We propose three strategies—fire survival, fast growth and affinity for the post-fire environment—that may confer a post-fire advantage to individual bacterial taxa and roughly map to stress-tolerant, ruderal and competitive strategies, respectively, although there are also important divergences from the original strategies as posited by Grime.

First, for the purposes of this paper, we define disturbance sensu ref. [12], that is, 'causal events that either (1) alter the immediate environment and have possible repercussions for a community or (2) directly alter a community'[13,14]. We note that this is a broader definition than that which Grime used, which was limited to processes that 'limit the plant biomass by causing its destruction'[10]. Disturbances can be classified as pulses or presses, on the basis of the period of disturbance[15]. Pulse disturbances are characterized by intense yet short-lived environmental pressures. Press disturbances can arise rapidly but persist over a longer period of time. The immediate effects of heat and combustion during wildfires can be considered as a pulse disturbance, while longer-term changes to vegetation communities and soil properties, such as pH and C availability[16], may be classified as press disturbances for soil microbial communities.

During pulse disturbances of wildfires, soil bacterial survival is affected by temperature and duration of heating, generally decreasing with both[17,18]. Thus, fire survival, conferred by strategies such as dormancy and potentially high GC content and heat-shock proteins, may be an important trait for maintaining bacterial populations, conceptually equivalent to the importance of seed banks for plant adaptations to wildfire[12,19]. For example, dormant cells have been shown to survive temperatures greater than 100 °C for 24 h in dry conditions[20]. While these survival strategies map roughly onto Grime's stress tolerance traits, stress tolerance as conceived by Grime is generally related to persistent stresses, including alpine, arid, shaded and low-nutrient environments[10]. In addition, the killing effect of a pulsed stress on the microbial community may also support ruderal or competitive strategies[10].

The press disturbance of wildfires on soil bacterial communities lingers long after the flames have receded and soil temperatures have returned to normal levels. Heat-induced bacterial mortality may open a niche for fast-growing taxa to recolonize. This is supported by previous work where higher 16S ribosomal RNA (rRNA) gene copy numbers were correlated with faster growth rates[21] and higher 16S rRNA gene copy numbers were observed at 4 months than at 2.5 yr post-burn[22]. Fast-growing bacteria with higher rRNA gene copy numbers may have a selective advantage in responding to increased nutrient availability post-fire[22]. Fire is named as a specific example of disturbance by Grime[10] and is associated with the ruderal strategy, with fast growth offered as a consistent plant trait characteristic of this strategy. Here, the pulse aspect of fire disturbance results in the rapid removal of organisms from the environment, creating space that fast-growing taxa fill, while the press aspect of fire disturbance determines the conditions under which ruderal or fast-growing taxa may flourish.

Press disturbances of burn-induced changes in soil physical and chemical properties may also advantage bacteria adapted to post-fire conditions. For example, soil pH, which has a strong influence on soil bacterial community composition[23], often increases with burning and elevated pH has been reported 4–9 yr post-fire[16,24]. In addition, pyrogenic OM, produced via the incomplete combustion of soil organic C[25],

has been shown to affect soil microbial community composition[26], although microbial community response is not consistent across varying soil types and chemical composition of fire-affected OM[27]. Increased pH, alteration of OM and/or other post-fire physicochemical changes may select for bacteria with an affinity for the post-fire environment. This trait diverges most from Grime's framework—while competition as conceived by Grime clearly takes place in the context of the specific environmental conditions, we are specifically concerned with 'differences in capacity to exploit (or in susceptibility to) features of the physical or biotic environment'[10], rather than competition per se.

Our objectives for this study were first to assess the relative importance of three fire response strategies—fire survival, fast growth and an affinity for the post-fire soil environment—in structuring the post-fire soil bacterial community over time (while recognizing that these traits would not necessarily be mutually exclusive within a given organism). We hypothesized that the importance of fire survival as a trait structuring bacterial communities would decrease with time since the fire, while the importance of fast growth in structuring bacterial communities would supersede fire survival as fast-growing bacteria recolonize the soil. In the years following a fire, an affinity for the post-fire soil environment would emerge as the most important strategy structuring bacterial communities. We hypothesized that each of these strategies would have greater effects on bacterial community composition following burns of higher severity[28]. Second, to probe the potential ecosystem-level importance of changes in bacterial communities post-fire, we explored whether fire-induced changes in bacterial community composition affected C cycling. We used experimental burns, soil incubations, and 16S DNA and rRNA sequencing, paired with a large field dataset 1 and 5 yr post-wildfire, to develop and test a traits-based fire ecology framework for soil microbes.

While ecological predictions can be made on the basis of the genetic features of a given organism or community, the extraordinary diversity of soil bacteria[29], the vast majority of which remain uncultured[30], and phenomena such as horizontal gene transfer impair our ability to use taxonomy or even genetics alone to confidently infer bacterial traits[31]. Thus, we used an uncommon approach to assign traits to bacteria in a high-throughput manner in this study (Fig. 1), where we first explicitly determined which individual bacterial taxa are able to survive fires, can grow quickly and/or are well-adapted to the post-fire environment. To identify traits, we worked with soil cores collected from 19 sites (Supplementary Table 1) in the boreal forest of Canada that had not burned in the previous 30 yr or more, using a series of experiments with simulated burns (Extended Data Fig. 1) and subsequent soil incubations. Fire survival was assessed by determining which bacterial taxa were enriched in burned vs unburned soil 24 h post-burn. Fast growth was assessed by determining which bacterial taxa became enriched after incubating soils for 5 weeks. Post-fire environment affinity was assessed by determining which taxa became enriched in burned soils vs unburned controls after a 6-month incubation. Then, we applied these trait assignments to a field dataset of natural wildfires from the same region, 1 and 5 yr post-burn[6,32], to evaluate the importance of each trait in the field. Finally, we used respiration data from the incubations of the experimentally burned cores to explore whether changes in microbial communities constrain soil C mineralization.

## Results

### Effects of laboratory burns on soil properties

We designed our fire simulations to span a range of fire effects while maintaining soil conditions that were representative of field conditions and a heat flux profile representative of natural wildfires. For each site, one core represented moist conditions, one core represented dry conditions and one control core was not burned. In the dry soil burns, fire propagated downwards from the soil surface, with maximum thermocouple temperatures reaching upwards of 600 °C at the organic–mineral interface (Fig. 2a and Supplementary Table 2). In the moist

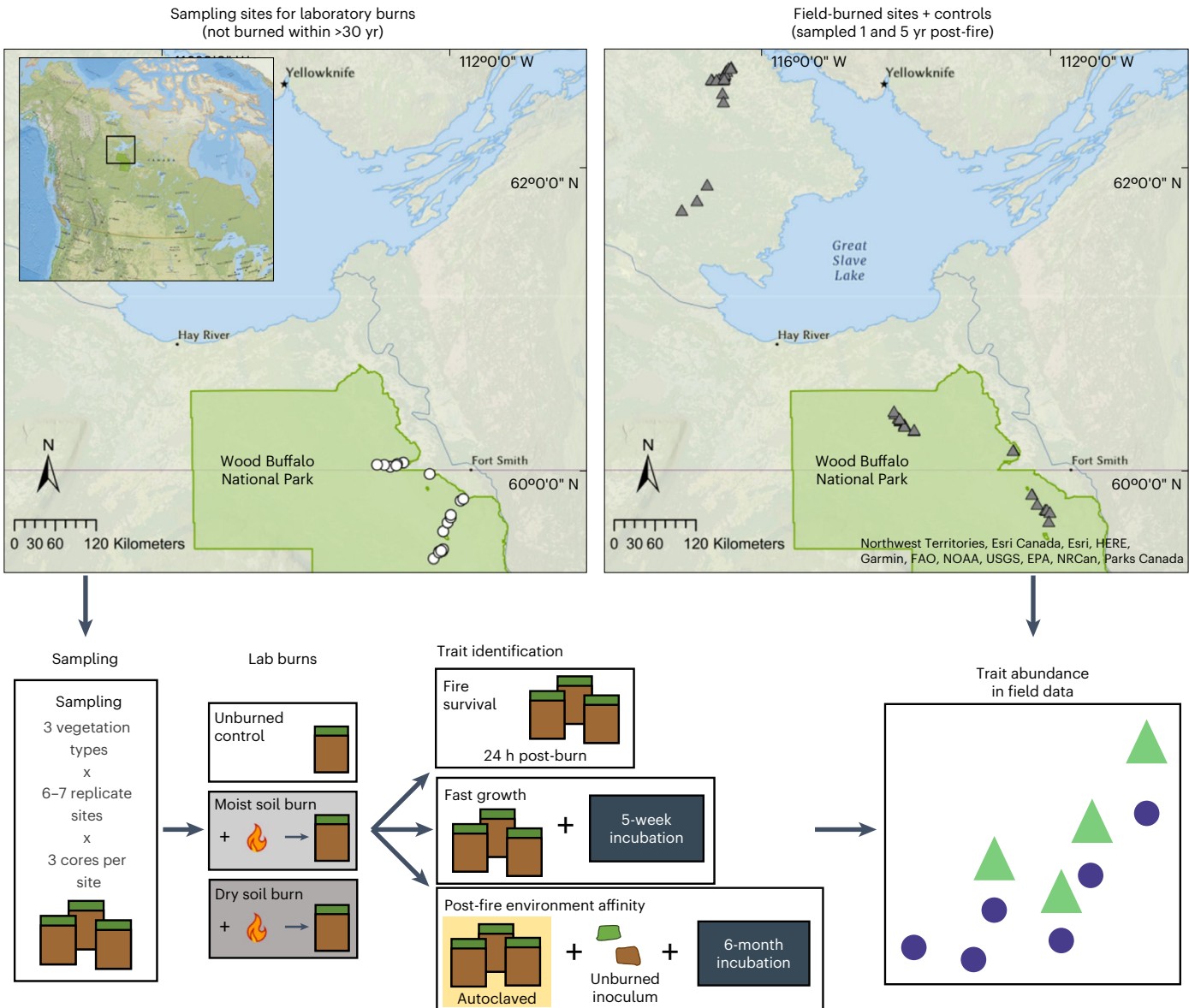

**Fig. 1 | Experimental design.** Sampling locations for laboratory burn experiments within Wood Buffalo National Park (WBNP), Canada (top left) and field-burned sites and controls in WBNP and the southern Northwest Territories (top right). Map created using ArcGIS Pro 2.7.0 (refs. 97,98). Inset indicates the relative location of the sampling region within North America. Simplified laboratory experimental design to identify bacterial taxa with fire-related traits (bottom left) and quantify the relative abundance of these traits in field data (bottom right).

soil burns, the thermocouple temperatures were much lower (maximum recorded is 42 °C). Soil pH was much higher after dry soil burns in organic horizons (mean = 7.2 ± 1.0) than in the unburned control organic horizons (mean = 4.5 ± 0.7; analysis of variance (ANOVA) and Tukey's honest significant difference (Tukey's HSD), $F$ = 82.7, d.f. = 1,34; $P$ < 0.001) (Fig. 2b and Supplementary Table 3). C:N ratios were much lower in organic horizons after dry soil burns (mean = 12.6 ± 4.3) than in the unburned (mean = 20.6 ± 5.9; ANOVA and Tukey's HSD, $F$ = 19.9, d.f. = 1,34; $P$ < 0.001) or moist soil burns (mean = 19.9 ± 4.7; ANOVA and Tukey's HSD, $F$ = 20.6, d.f. = 1,34; $P$ < 0.001) (Fig. 2b and Supplementary Table 3). Decreases in C:N were largely driven by decreased total C concentration; organic horizons subjected to the dry soil burns had a mean decrease in C concentration of 11.7% (ANOVA and Tukey's HSD, $F$ = 10.98, d.f. = 1,34; $P$ = 0.002) compared with unburned soil, whereas mean total N concentration was not significantly affected by burn treatment in either organic or mineral horizons (Fig. 2b, Extended Data Fig. 2 and Supplementary Table 3).

## Impact of laboratory burns on soil microbial respiration

Two-pool exponential decay models resulted in good fits to respiration data for all samples ($R^2$ of 0.98–1, $P$ < 0.0001) (Extended Data Figs. 3 and 4). In the fast-growth incubation, the fractional size of the fast C pool ($M_1$) was significantly smaller in the burned cores (mean = 0.027 ± 0.019) than in the unburned cores (mean = 0.096 ± 0.10; Wilcoxon signed-rank test, $P$ = 0.001) (Fig. 2c, and Supplementary Tables 4 and 5). The fast C pool decay rate coefficient ($k_1$) was higher in the burned cores (mean = 0.16 ± 0.096) than in the unburned cores (mean = 0.0036 ± 0.015; Wilcoxon signed-rank test, $P$ < 0.001). The slow C pool decay rate coefficient ($k_2$) was lower in the dry soil burn (mean = 0.00098 ± 0.00072) than in the unburned soil (mean = 0.0013 ± 0.00086; Wilcoxon signed-rank test, $P$ = 0.00002) and in the moist soil burns (mean = 0.0017 ± 0.00085; Wilcoxon signed-rank test, $P$ = 0.003). C flux results for the post-fire environment-affinity incubation were similar (Supplementary Tables 4 and 5, and Extended Data Fig. 5).

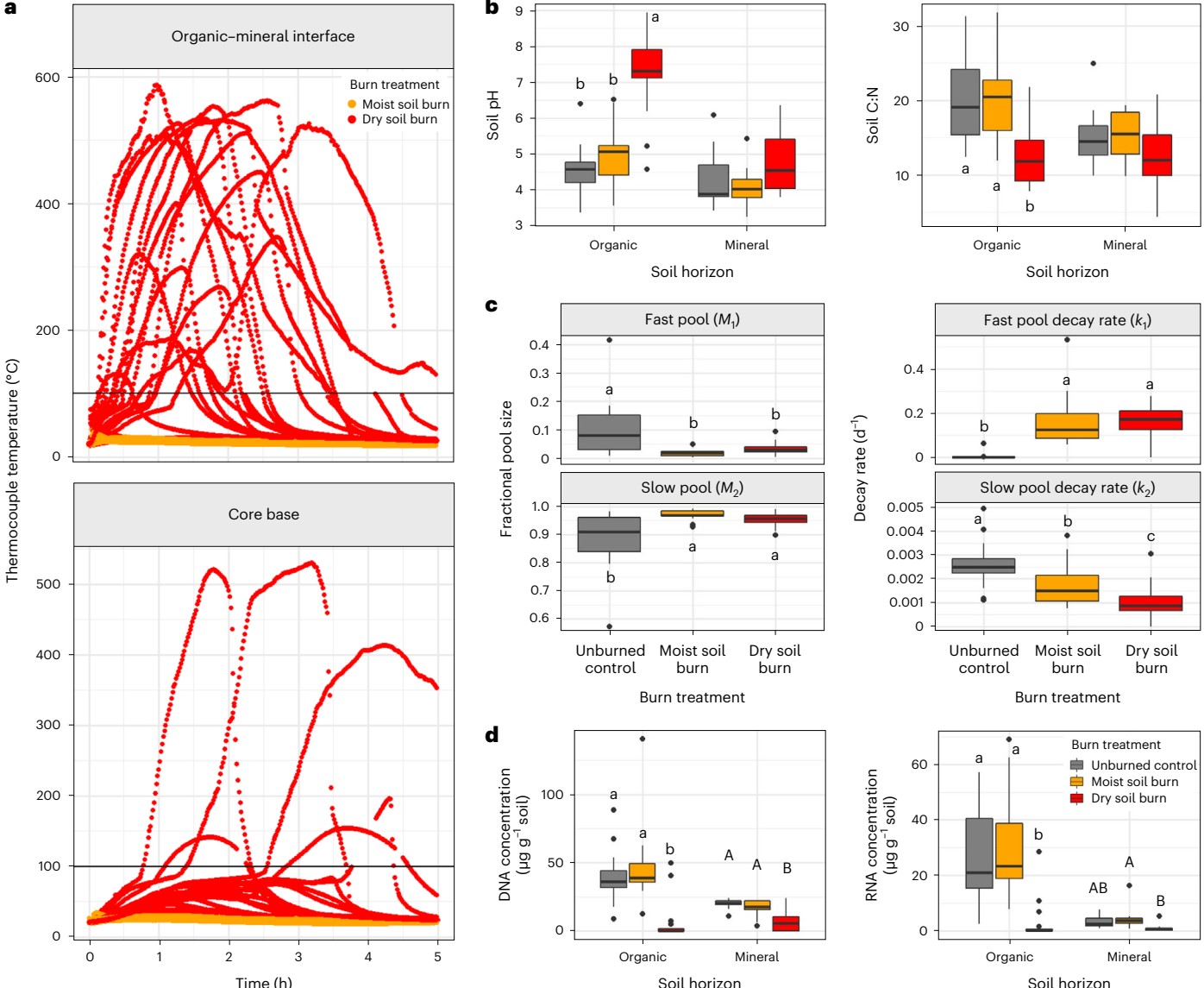

**Fig. 2 | Fire effects on lab-burned soil cores. a**, Thermocouple temperatures at the organic–mineral soil interface (top panel) and 1 cm above the core base (bottom panel) during the 5 h following the initiation of the burn treatments ($n = 19$ moist soil burn, $n = 19$ dry soil burn). Burn simulations were initiated at time 0. Temperatures greater than 100 °C suggest the presence of flaming combustion and are not a quantitative measure of soil matrix. **b**, Soil pH (left) and C:N (right) across soil horizons $n = 18$ for all O horizons, $n = 10$ for unburned mineral horizons, $n = 11$ for moist burn mineral horizons and $n = 12$ for dry burn mineral horizons. **c**, Coefficients for two-pool exponential decay models fit to soil C respiration data ($R^2$ of models 0.98–1, $P < 0.0001$; $n = 18$ for unburned and moist soil burn cores and $n = 19$ for dry soil burn cores). Fractional size of the fast ($M_1$) and slow ($M_2$) C pool (left two panels) and decay rate coefficients for the fast ($k_1$) and slow ($k_2$) C pool (right two panels) across burn treatments for the 5-week fast-growth incubation. **d**, Concentration of total DNA (left) and RNA (right) extracted from soil 24 h post-burn ($n = 19$ for O horizons for all burn treatments, $n = 11$ for unburned mineral horizons, $n = 12$ for moist burn mineral horizons and $n = 13$ for dry burn mineral horizons). For all boxplots, different letters represent statistically significant treatment differences based on ANOVA and Tukey's HSD ($P < 0.05$). The central horizontal line indicates the median, the upper and lower bounds of the box indicate the inter-quartile range (IQR), the upper and lower whiskers reach the largest or smallest values within a maximum of 1.5 × IQR and data beyond the whiskers are indicated as individual points.

## Community composition in laboratory experiments and in the field dataset

Within the laboratory data, for all communities across the three trait identification experiments, all tested factors (dominant vegetation, soil horizon, pre-burn horizon thickness, pH, total C and N, soil texture and burn treatment) were significant predictors of community composition in combined models (permutational multivariate ANOVA (PERMANOVA), $P < 0.05$ for all factors) (Extended Data Fig. 6 and Supplementary Tables 6–9; total operational taxonomic units (OTUs) per sample reported in Table 10). Burning strongly decreased total RNA (10×) and DNA (7×) concentrations for dry soil burns in the organic

horizon (Fig. 2d; ANOVA and Tukey's HSD, $P = 6×10^{-8}$) and decreased DNA concentrations (3×) in the mineral horizon (Fig. 2d; ANOVA and Tukey's HSD, $P = 1×10^{-5}$). In organic horizons, community compositions changed more with burning at the sites where pH increased more after dry soil burns ($R^2_{adj} = 0.5$; $P = 0.0008$; Supplementary Fig. 1). Differences in weighted mean predicted 16S rRNA gene copy number emerged after 5 weeks of incubation: dry soil burns had significantly higher weighted mean predicted rRNA gene copy number (mean = 3.4 ± 0.9, organic; mean = 2.7 ± 0.9, mineral) than unburned soil (mean = 2.0 ± 0.4, organic; mean = 1.7 ± 0.1, mineral; ANOVA and Tukey's HSD, $P < 0.0001$).

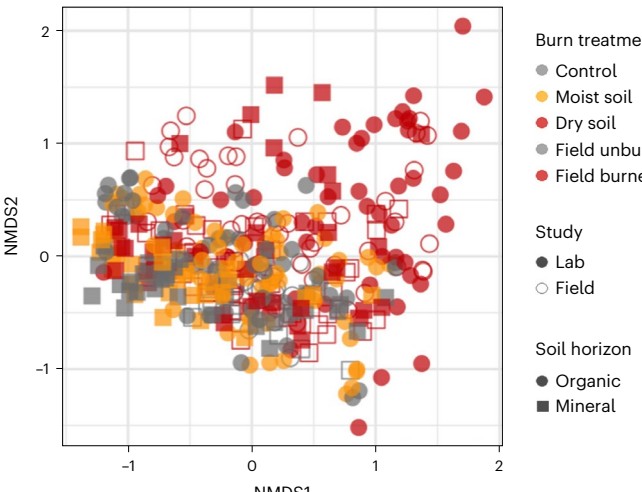

**Fig. 3 | Community composition of laboratory and field samples.** First two axes of non-metric multidimensional scaling (NMDS) ordination of Bray−Curtis[99] dissimilarities for bacterial communities from lab study samples (24 h incubation, DNA only; 5-week incubation; 6-month autoclaved post-fire-affinity incubation) and field samples (1 and 5 yr post-fire) ($n$ = 412 samples; $k$ = 3, stress = 0.13).

There was good correspondence in overall community composition between our lab samples and our field samples (Fig. 3), with taxa detected in the lab samples representing a mean of 93% of total reads (minimum 70%, maximum 99%) in the field samples, suggesting that the laboratory samples were not unrealistically altered through our experimental manipulations. In addition, for the post-fire environment-affinity experiment, the autoclaving treatment provided the least explanatory power for community composition ($R^2$ = 0.02; Supplementary Table 9), suggesting that soil properties related to burning played a larger role in structuring communities than any artefacts of autoclaving before inoculation.

### Fire survival in the lab and in the field
We identified 20 fire survivor OTUs (Supplementary Tables 11–13). These fire survivor taxa were significantly positively correlated with burn severity index in the field samples ($P$ = 0.00005, $R^2_{adj}$ = 0.14), both 1 and 5 yr post-fire (interaction term between years post-fire and burn severity index, $P$ = 0.50), although they made up a relatively small fraction of the total community (Fig. 4a).

### Fast growth in the lab and in the field
We identified 71 fast-grower OTUs (Supplementary Tables 11–13). Of these fast-grower taxa, two were also identified as fire surviving. These fast-growing taxa were significantly positively correlated with burn severity index ($P$ = 1 × 10⁻⁸, $R^2_{adj}$ = 0.34; Fig. 4b) and represented a sub-stantial fraction of the total reads (mean = 14 ± 11%) 1 yr post-fire in high-severity burns (burn severity index ≥3). However, this relationship was completely gone 5 yr post-fire (interaction term between years post-fire and burn severity index, $P$ = 0.00006), with fast growers rep-resenting only a mean of 2 ± 1% of total reads in high-severity burns (burn severity index ≥3).

There were two potential confounding factors in the significant positive relationship between relative abundance of lab-identified fast growers and burn severity in field samples 1 yr post-fire that we also wanted to control for. First, because fast-growing taxa often have higher 16S rRNA gene copy numbers, we wanted to ensure the relation-shp was not an artefact of this phenomenon. After normalizing relative abundances by predicted 16S rRNA copy number for each OTU[33], the relationship remained significant ($P$ = 0.00008, $R^2_{adj}$ = 0.20). Second,

because total community size could also vary with burn severity, we combined relative abundances with quantitative polymerase chain reaction data published in the original field manuscript to provide estimates of absolute abundances, after which the relationship also remained significant ($P$ = 0.02, $R^2_{adj}$ = 0.07). The relative abundance of lab-identified fast-growing taxa, adjusted for predicted rRNA copy numbers to reduce potential spurious correlations, was significantly positively correlated with the community-level weighted mean pre-dicted rRNA gene copy number (Extended Data Fig. 7; $P$ = 2 × 10⁻¹⁵, $R^2_{adj}$ = 0.62).

### Post-fire environment affinity in the lab and in the field
We identified 31 post-fire environment-affinity OTUs (Supplementary Tables 11–13). Of these taxa, one was also identified as fire surviving and two were identified as fast growing. These taxa with affinity for the post-fire environment were significantly positively correlated with burn severity index in the field samples, although they made up a relatively small fraction of the total community ($P$ = 0.001, $R^2_{adj}$ = 0.11; Fig. 4c). This relationship was weaker 5 yr post-fire than 1 yr post-fire (interac-tion term between years post-fire and burn severity index $P$ = 0.06).

### Fire traits correlate with community shifts after burning
Total lab-identified responders represented up to 44% of the field-burned communities (mean = 17 ± 11% 1 yr post-fire for burn sever-ity index ≥3). Total lab-identified responders were also significantly correlated with Bray−Curtis dissimilarity of burned sites vs unburned sites, matched for vegetation, soil horizon and year ($P$ = 7 × 10⁻⁷, $R^2_{adj}$ = 0.37; Fig. 5), with this relationship weakening between 1 and 5 yr post-fire ($P_{interaction}$ = 0.003).

## Discussion
Our experimental burns directly killed microbes (Fig. 2d) and also altered soil physical and chemical properties (Fig. 2b), which, together, caused significant changes to bacterial communities that were char-acteristic of natural wildfires in this region of the boreal forest (Fig. 3) and affected microbial C mineralization (Fig. 2c).

### Importance of fire survival is small but persistent
Our hypothesis that fire survival would be most important in structur-ing bacterial communities following higher-severity burns was gener-ally supported by our data, but our prediction that its influence would diminish over time was not (Fig. 4a). Unsurprisingly, burning caused a decrease in total RNA and DNA concentrations in the soil (Fig. 2d), which can be explained by bacterial (and other organismal) mortality, along with destruction of necromass and relic DNA[34]. However, our approach to identifying fire survivors yielded only a very small number of taxa as fire survivors (20 of the 19,083 total OTUs across the laboratory genomic DNA dataset), suggesting, somewhat counter-intuitively and in contrast to many plant species[35], that fire survival is not an important strategy for bacteria within a year post-fire.

How can that be? We posit that there is probably a relatively narrow optimal zone[36] for fire temperatures where survival plays a meaningful role. At sufficiently high burn temperatures, there is probably almost complete microbial death[17], making differential survival irrelevant. However, at low burn temperatures, characteristic of moist soil burns and deeper soil horizons (Fig. 2a), there is very little microbial death (Fig. 2d), so survival is again not a relevant strategy. This observa-tion is supported by the field data, where the sites with the greatest proportion of fire survivors were those with intermediate severity burns (Fig. 4a).

Shorter-term observations taken closer to the 'pulse' timescale of fire disturbance may reveal a greater role for fire survival, but in the field, any short-term effects of fire survival may also be rapidly over-whelmed by dispersal, which can increase soil bacterial community resilience[37]. In boreal forests, spatial heterogeneity in fire severity

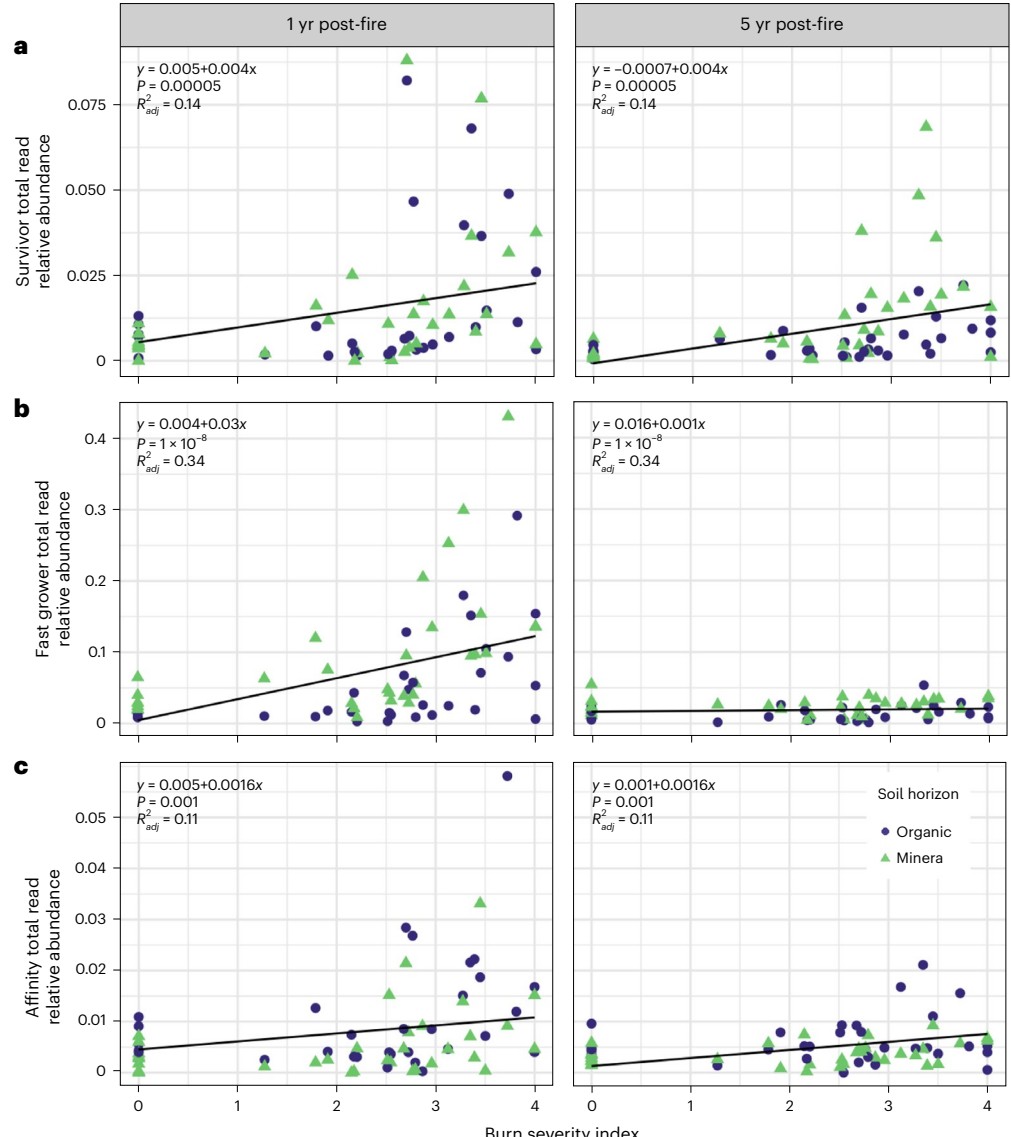

**Fig. 4 | Prevalence of lab-identified traits in field community data 1 and 5 yr post-fire. a–c**, Total relative abundance of lab-identified (**a**) fire-surviving, (**b**) fast-growing and (**c**) post-fire environment-affinity taxa in field data at 1 yr (left) and 5 yr (right) post-fire vs burn severity index ($n$ = 66 field soil samples). Linear models including interaction terms between years since the fire and burn severity index were used to test for a significant correlation between burn severity and total relative abundance of all taxa within a given trait. If the interaction was not significant, results from the model are reported without the interaction.

results in complete combustion of soil organic horizons in some areas while adjacent 'fire refugia' remain unburned[38]. Similarly, even when lethal burn temperatures are reached at the surface, temperatures rapidly attenuate with depth (Fig. 2a), supporting belowground fire refugia from which dispersal to surface horizons could occur as well. Thus, we propose that fire survival can be an important strategy structuring post-fire microbial communities for certain fires and timescales, but only when a narrow range of conditions is met: (1) temperatures reach levels that are lethal for some taxa, but—critically—not for others; (2) post-fire dispersal from above or below is not sufficient to overwhelm this effect.

For the few taxa identified as fire survivors, what traits allow them to survive? Fire survival may be associated with dormancy[39] and may translate readily to the ecological function of seed banks for plants in fire-prone ecosystems[40]. Some *Actinobacteria* and *Firmicutes* that were enriched following high-severity burns have been found to possess sporulation genes, suggesting that sporulation and dormancy may aid in fire survival[19,41]. Two of our fire survivors were classified as *Bacillus* sp. (Supplementary Table 11), which is a known endospore-forming genus[42] and, combined, accounted for as much as 13% of total reads in dry soil burns immediately post-fire in laboratory experiments. Taxa such as these may survive burning in a dormant state and then either germinate and grow, or persist in a dormant state after the fire. Some taxa enriched following high-severity burns have also been found to possess genes encoding heat-shock proteins and/or a higher GC content, both of which may be additional traits promoting fire survival[19]. Two of the fire survivors (from the genera *Domibacillus* and *Clostridium*) were also identified as being fast growers (Supplementary Table 11), highlighting potential positive interactions between strategies (analogous to Grime's 'stress-tolerant ruderals'[10]) and the possibility of dormant cells surviving and germinating post-fire and then rapidly colonizing the burned soil. However, fire survival was a less common strategy than fast growth, and the traits promoting fire survival, such as dormancy and heat-shock proteins, would not necessarily be expected to directly promote fast growth.

### Importance of fast growth is large but short-lived
We hypothesized that, with time, the importance of fire survival would be superseded by fast growth, as fast growth would enable bacteria to take

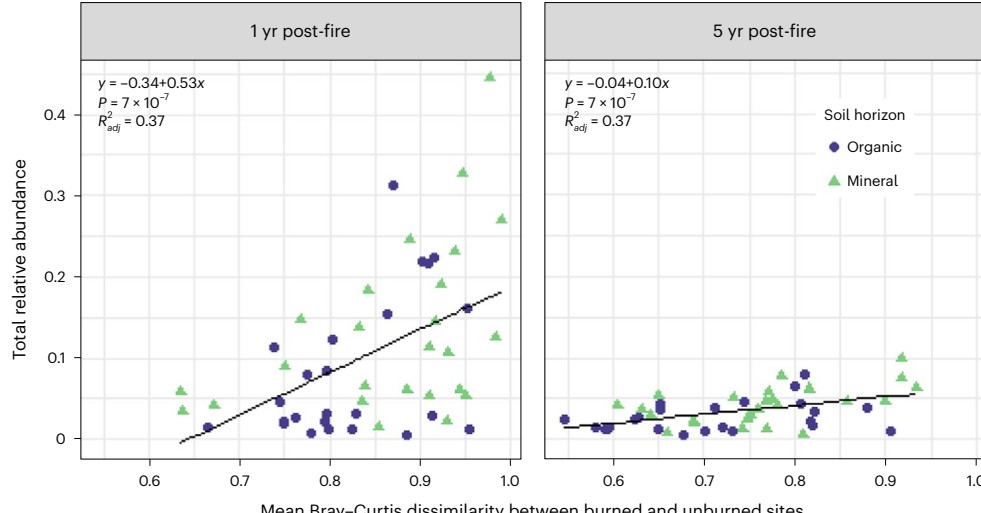

**Fig. 5 | Relationship between lab-identified traits and shifts in community composition with fire in the field.** Total relative abundance of all taxa with lab-identified fire-related traits in field data 1 yr (left) and 5 yr (right) post-fire vs mean Bray–Curtis dissimilarities between burned and all corresponding unburned field sites (matched by vegetation community, soil horizon and year). Lines indicate linear models ($n = 49$ field soil samples, $P = 7 \times 10^{-7}$, $R^2_{adj} = 0.37$) with the following equations: 1 yr post-fire: $y = -0.34 + 0.53x$; 5 yr post-fire: $y = -0.04 + 0.10x$.

advantage of an increase in available nutrients released via burning and to rapidly colonize the burned environment. Our results were generally consistent with this hypothesis, where these fast-growing taxa accounted for a large proportion of total reads and were positively correlated with burn severity 1 yr post-fire (Fig. 4b), further indicating that in the short-term, fast growth plays a strong role in structuring community composition. This rapid increase in fast-growing taxa is probably fuelled by nutrients liberated by fire via heat-induced bacterial mortality, vegetation death and pH increase, combined with reduced competition[43]. However, the positive correlation between burn severity and relative abundance of fast growers completely disappears by 5 yr post-fire. Despite this, microbial communities can take longer than 5 yr to recover post-fire (Fig. 3)[44], indicating that strategies other than fast growth probably play key roles in structuring soil communities over longer timescales.

Our results also lend clear support to the interpretation of weighted mean predicted 16S copy number as a metric related to fast growth potential in environmental samples (Extended Data Fig. 7), given the strong correlation ($R^2_{adj} = 0.62$) between this metric and the relative abundance of lab-identified fast-growing taxa. While this metric is often interpreted in this context[6,32], there are not many field studies empirically supporting this interpretation. The lab-burned soils, when considered on their own, also followed these trends: by 5 weeks post-burn, burned communities had higher weighted predicted mean 16S rRNA gene copy numbers than unburned controls, indicating that fast growth quickly becomes important following burning.

Several of the fast growers, such as *Arthrobacter* sp. and *Massilia* sp., have been identified as being enriched post-fire in previous studies[6,32,45] and accounted for up to 38% of total reads in field data. Upon first consideration, these fast growers may seem ecologically analogous to canonical ruderal, fast-growing deciduous trees that are characteristic of the boreal forest, such as trembling aspen[46]. However, high-severity fires in the region and shorter fire return intervals have been linked to a persistent shift in dominance away from conifers towards aspen[47]. In contrast, we observed that for bacteria, evidence for fast growth as a relevant trait was essentially absent 5 yr post-fire. This contrast may be explained by differences in co-occurring strategies between plants and microbes. While numerous factors interact to shift plant communities towards aspen under high-severity fires, including increased bare mineral seed beds, survival via underground structures and rapid resprouting post-fire[48], we identified few bacterial

taxa with more than one fire-related trait. It is also probably important to consider the dramatically different lifespans and population turnover times for bacterial communities vs those of plants; while individual bacterial cells can certainly persist, dormant in the soil for decades or longer, turnover times of less than a day have also been observed for some soil bacteria[49], that is, the timescale of a pulse vs a press disturbance is probably perceived very differently by plants and bacteria. Due to the extremely small volume of soil that an individual cell must rely on for its habitat and sustenance, specific soil conditions may also play a comparatively outsized role in determining bacterial community composition, while plants, via their roots, can explore a much larger volume of the soil environment. Finally, the relative importance of biogeography and community assembly processes, such as dispersal or the richness of adjacent source populations, would certainly be expected to differ between bacteria and plants, although these phenomena remain under-characterized for bacteria[50,51].

Given the large fingerprint of fast growers post-fire, we wanted to ask: are there important implications for ecosystem functioning due to this bloom of fast-growing bacteria? For the first 2 weeks, respiration rates (per gram initial soil C) in the incubated soils were equivalent or higher in dry burned soil vs unburned soil (Extended Data Fig. 8a). This suggests that any limitation to mineralization due to high initial bacterial mortality (Fig. 2d) was rapidly offset by fast-growing taxa. However, by 3 weeks post-burn, respiration rates (per gram initial soil C) were lower in dry burned soil than in unburned soil (Extended Data Fig. 8a). This reversal in trend could indicate that microbes rapidly depleted the already smaller fast C pool, shifting to greater reliance on the larger but more persistent fire-altered slow C pool (Fig. 2c).

### Importance of post-fire environment affinity is short-lived
We hypothesized that over the years following a fire, an affinity for the post-fire soil environment would emerge as the most important strategy structuring bacterial communities. However, the importance of this strategy observed in the field was small, albeit statistically significant (Fig. 4c). This may be due in part to the fact that 'affinity for the post-fire soil environment' probably encompasses myriad traits that depend on specific soil properties—that is, there may be some properties that tend to be common across burned soils.

Physicochemical changes to soil properties, such as pH, post-fire were often extreme (Fig. 2b) and probably constrained post-fire

community composition, offering opportunities for bacteria with adaptive traits to thrive. The laboratory soils where the community changed the most with burning also had the greatest increases in pH (Supplementary Fig. 1), which could help explain why reintroduction of bacteria from unburned soils had negligible effects on community composition (Extended Data Fig. 6).

Along with pH shifts, post-fire environment affinity could be conferred by the ability to decompose OM that is altered during the fire. While we did not characterize chemical composition of soil OM directly, all dry soil burns resulted in some degree of organic horizon combustion, and temperatures during the dry dry soil burns would have been high enough to chemically alter OM[52]. This is reflected in the increase in the fractional size and decrease in decay rate of the slow C pool of the burned soils (Fig. 2c). The ability to degrade fire-altered OM may become increasingly important after easily mineralizable fire-liberated C, such as dead microbes and fine roots, is depleted, but before fresh inputs from vegetation have returned to pre-fire levels. Supporting this suggestion, in the field data, the importance of post-fire environment affinity taxa was positively correlated with burn severity (Fig. 4c) and this relationship weakened 5 yr post-fire, which could be driven by fresh inputs to fast C pools as boreal plant communities rebounded[47]. Thus, for determining microbial community composition, we propose that the immediate effects of fire on soil properties, such as pH shifts or losses and transformations of soil OM, may be less important over time than the effects on soil conditions mediated by plant community re-establishment that emerge over time, such as changes to C inputs, water and nutrient availability, symbioses and microclimatic effects[53].

## Fire response traits at the community level

Our experimentally based approach to identifying traits captured taxa representing up to 60% of total reads (mean = 15 ± 14% in dry burns) in laboratory-burned samples (Extended Data Fig. 9) and up to 44% in field data (mean = 17 ± 11% in sites with burn severity index ≥3 1 yr post-fire). However, most responsive taxa (96%) were only assigned to one trait. On the one hand, this paucity of OTUs associated with two or more traits could offer evidence for a trait trade-off. However, while Grime's original framework does invoke a trade-off between strategies[10], the traits we have focused on here do not map perfectly to that original framework and may not be expected to require trade-offs. We purposefully took a conservative approach to identifying responsive taxa and note that there is also evidence that some fire-relevant traits may be complementary. For example, *Bacillus* sp. are often noted both for their ability to form stress-resistant endospores as well as for their high 16S rRNA gene copy number and potential for fast growth[54], and were among our tandem fire survivors and fast growers. Further work will be needed to fully explore the prevalence of strategy co-occurrences or trade-offs post-disturbance.

The significant positive relationship between how different burned communities were from unburned communities and the total relative abundance of taxa with any of the three fire-related traits (Fig. 5) further underscores that we were able to identify critical fire-related traits, as the communities that were most altered by fire overall also had the greatest fraction of our lab-identified taxa. It also suggests that fire-related bacterial traits are consistent with community-level resilience: even at sites with high-severity burns, large changes in community composition and high prevalence of fire-related bacterial traits, bacterial communities became more similar to unburned communities 5 yr post-fire[32]. It will be interesting to continue to investigate how changing fire regimes in the boreal forest affect these dynamics. For example, while increasing fire frequencies and changes in burn severity in plant communities in this region have resulted in community shifts towards some fire-adapted plant taxa (for example, aspen) over others[47], analogous shifts in bacterial community composition appear to be more subtle so far[55].

## Post-fire C cycling is not constrained by bacterial community

The laboratory burn-induced changes to the chemical environment (Fig. 2b) were typical of wildfires: at high temperatures, the complete combustion of organic C and formation of mineral ash have been observed to increase soil pH by as much as 3 units[56]. The decreases in C:N we observed in organic horizons are probably explained by the lower temperature thresholds for C volatilization than for N: during combustion of OM, C losses begin around 100 °C, whereas N volatilization begins around 200 °C[57]. Although narrower C:N ratios are often thought of as a proxy for higher OM lability, accompanying chemical changes to soil OM during combustion and pyrogenic OM production may override this trend[58,59].

While total C was lost during combustion (Extended Data Fig. 2), resulting in lower respiration rates on a dry soil mass basis (Extended Data Fig. 8b), fire-induced changes in soil properties, combined with direct and indirect effects of fire on microbial communities, also affected how the remaining C was mineralized (Fig. 2c). The decrease in fractional size of the fast C pools with burning (Fig. 2c) is consistent with loss of easily mineralized OM via combustion. The increase in the decay rate of the fast C pool with burning despite the decrease in its fractional size (Fig. 2c) could be explained by a change in the chemical composition of the fast C pool, such as the production of necromass and increased oxidation of the remaining OM[57]. The increase in fractional size and decrease in decay rate of the slow C pools (Fig. 2c) with burning is consistent with pyrolysis of OM during burning leading to production of pyrogenic OM, which is less readily decomposed by microbes[60,61].

We infer that the observed changes in C mineralization were probably driven primarily by changes in the soil environment and changes to OM chemistry and stocks, rather than by a loss of functional potential in the microbial community. The evidence supporting this interpretation is that inoculating the soil with living microbes did not affect mineralization rates: the C fluxes in the uninoculated 5-week fast-growth incubation were indistinguishable from those in the first 5 weeks of the inoculated (post-fire environment affinity) incubations (Extended Data Fig. 10). If the microbial community had been significantly impaired with respect to its ability to mineralize C due to either changes in composition or persistent reductions in biomass, we would expect inoculation to increase C mineralization rates, which we did not observe. Microbes can rapidly grow in the post-fire environment, as evidenced by the rapid proliferation of fast growers (Fig. 4b). Together, these suggest that despite large shifts in bacterial community composition and size post-fire, strong functional redundancy and ability for fast growth post-fire maintain the microbial community's collective ability to mineralize OM. Future studies translating these findings to field settings in the boreal forest would help confirm these effects and the timescales over which they might occur.

## Trait-based framework

Drawing on our observations, we offer a hypothetical framework of response to fire for bacterial communities (Fig. 6). The importance of the three fire-adaptive traits presented here varies over time and with burn severity (or soil depth).

Immediately post-fire, some taxa are particularly strong survivors. However, this effect is primarily relevant for burns of intermediate severity and/or intermediate soil temperatures (~50 °C–100 °C), with lower-temperature burns leaving the community generally intact and higher-temperature burns wiping out all organisms. Under (rare) survivor-optimal burn conditions, fire-surviving taxa can remain enriched in the community for years after the fire (Fig. 4a). Fast-growing bacteria rapidly fill the niche created by bacterial mortality and a pulse of available C sources, and constitute a large portion of the total community. Higher-severity burns select for more fast-growing taxa due to greater overall mortality and a larger open niche space. However, the rapid proliferation of these taxa essentially disappears between 1 and 5 yr post-fire. Over the same time period, a smaller set of taxa with an

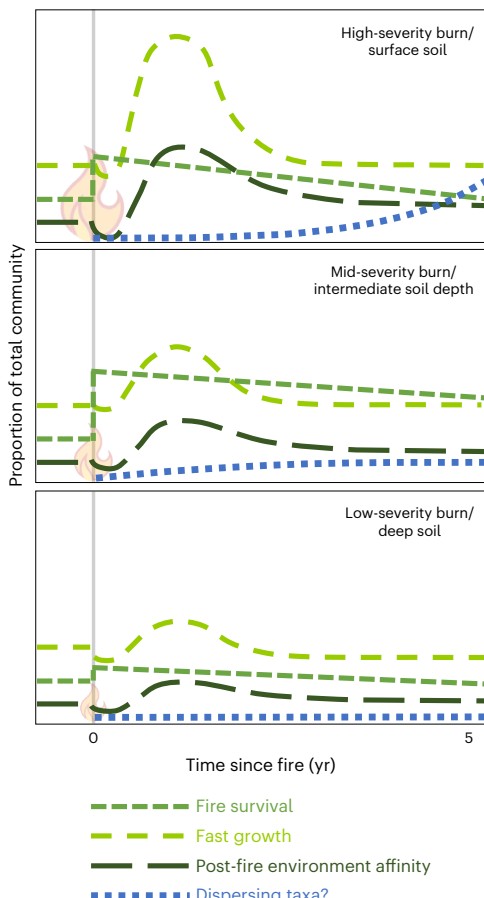

**Fig. 6 | Hypothetical traits-based fire ecology framework.** Panels indicate the relative importance of fire survival, fast growth, post-fire environment affinity and dispersal in structuring post-fire soil bacterial communities for high (top), mid (middle) and low (bottom) severity burns or soil depths. (Note that, while hypothesized fire survival, fast growth and post-fire environment-affinity trends are based directly on our observations in this study, our predictions for the importance of dispersing taxa are largely speculative.)

affinity for post-fire environmental conditions thrive, possibly via tolerating fire-induced changes in soil properties such as pH, or by specialization for fire-generated compounds. Due to above- and belowground fire refugia, dispersal limitation may not be a particularly important factor in structuring post-fire communities shortly after the fire, even for high-severity burns, and establishment of dispersing organisms may be constrained by fire-affected chemical properties. Over longer periods, additional strategies not characterized in this study, such as those related to the effects of recolonizing plant community on soil properties[53], may become increasingly relevant.

## Conclusion

Using experimental burns and laboratory incubations to directly and empirically identify and assign traits to fire-surviving taxa, fast-growing taxa and post-fire environment-affinity taxa, we were able to account for large fractions of the total community in field samples collected across a range of vegetation, soil types and wildfire characteristics. The functional implications of these fire-induced changes to microbial communities, as assessed by directly measuring $CO_2$ fluxes from the burned cores and fitting two-pool decay models, indicate that soil C losses post-wildfire are not likely limited by the microbial community—a finding with important implications for understanding global change and biogeochemical cycling post-fire. Our findings help expand our understanding of fire ecology beyond the better-characterized plant

world to soil bacteria, identifying analogies and contrasts, and we offer a hypothetical framework for the relevance of these post-fire strategies over time. Future work remains to explore the functional implications of these strategies, identify additional strategies not characterized in this study and evaluate relationships between recolonizing plant communities, soil properties and microbial strategies over decadal timelines. Next steps will also include testing this framework and identifying these taxa in other fire-adapted ecosystems. Finally, our integrated experimental–field observational approach could be translated and applied to many different systems.

## Methods

For further details on methods, please see Supplementary Information.

### Study overview, site description and sample collection

For the experimental burns, soil cores were collected from 19 sites across Wood Buffalo National Park (WBNP), Canada in June 2019 (Permit number WB-2019-31497; $n = 19$ sites, $n = 57$ organic horizons, $n = 36$ mineral horizons for all analyses unless otherwise stated; Fig. 1, Extended Data Fig. 1 and Supplementary Table 1). The park has both sandy acidic soils (Eutric Gleysols and Eutric Cambisols, FAO) and organic-rich peatlands (Dystric Histosols, FAO) underlain by discontinuous permafrost[62–64]. The dominant vegetation of the sampled regions of the park is jack pine (*Pinus banksiana* Lamb.), trembling aspen (*Populus tremuloides* Michx.), black spruce (*Picea mariana* (Mill.)) and white spruce (*Picea glauca* (Moench) Voss).

Sites that had not burned in the previous 30 yr or more were selected using stratified random sampling using the Canadian National Fire Database[65], dominant vegetation from the Canadian National Forest Inventory[66] and soil type from FAO soil survey data[67]. At each site, three soil cores (15.2 cm × 7.6 cm diameter) were collected across a 2 × 2 m grid. All cores had surface organic soil horizons and 36 of the 57 cores also contained a portion of underlying mineral soil horizon. Cores were stored at room temperature or cooler and transported to Madison, Wisconsin within 14 d of collection. Cores were then air dried for 6 weeks to simulate extreme dry conditions with no precipitation[68] and to allow standardization of moisture contents across sites.

### Fire simulations

We designed our fire simulations to span a range of fire effects while maintaining soil conditions that were representative of field conditions and a heat flux profile representative of natural wildfires. For each site, the three cores were used for two burn simulations and a control. The first core ('moist soil') was saturated and then allowed to free-drain for 24 h (that is, reaching field capacity) before the burn and represented moist conditions, while the second core ('dry soil') was not wet up before the burn and represented drought conditions. The third core was not wet up and not burned ('control'). Cores were exposed to a cumulative 7.2 MJ m$^{-2}$ radiant heat flux, typical of a mid- to high-range crown fire in this region[69], at 60 kW m$^{-2}$ for 2 min in a mass loss calorimeter (Fire Testing Technology) and allowed to cool overnight (Extended Data Fig. 1). Temperatures during and up to 12 h after the burn were measured with beaded 20-gauge Type K thermocouples (GG-K-20-SLE, Omega) 1 cm above the base of the core and at the transition between the organic horizon and the top mineral horizon or, for completely organic cores, 5 cm above the core base.

### Soil property analyses

At 24 h after the burn, mass loss was recorded (Supplementary Table 14). Cores were then subdivided by the organic-rich surface soil and the underlying mineral soil, where present. Subsamples of each horizon were immediately preserved for pH measurements[70] and C and N analysis (Flash EA 1112 CN Automatic Elemental Analyzer, Thermo Finnigan). Soil texture was determined using a physical analysis hydrometer at the UW-Madison Soil and Forage Lab (Supplementary Table 1).

## Soil incubations and gas flux tracing

After the burn simulations, we used three different experiments to assign the three traits (Fig. 1). Fire survival was assessed by determining which bacterial taxa were enriched in burned vs unburned soil 24 h post-burn, as described in more detail below. Fast growth was assessed by determining which bacterial taxa became enriched after incubating soils for 5 weeks. Post-fire environment affinity was assessed by determining which taxa became enriched in burned soils vs unburned controls after a 6-month incubation.

For the post-fire environment-affinity trait, we wanted to determine which bacteria were suited to the post-fire soil conditions, particularly, regardless of their ability to survive the fire. Our approach was designed to allow inclusion of taxa that might not survive the fires but may flourish under post-fire soil conditions if they were able to disperse to the burned soils. Thus, we inoculated burned soils with unburned soils to introduce living bacteria representative of unburned sites. To reduce existing total populations, we autoclaved soils before inoculating them with living microbes, adding unburned soils from the same site on a 10% dry mass basis by horizon[71,72]. To characterize any artefacts from autoclaving, we also included control incubations using soil from each core that was inoculated without having been autoclaved. Following inoculation, all soils were incubated for 6 months.

For all incubations, mineral soil from each sample (where present) was packed into a 60 ml glass jar and then topped with soil from the sample's organic horizon to recreate original soil profiles proportionally, packing them to the bulk density of the original core and maintaining them at optimal soil moisture conditions. $CO_2$ efflux was tracked over the course of all incubations using KOH traps[73].

## Nucleic acid extraction and sequencing

To assess fire survival potential, RNA and DNA were extracted from soil collected at 24 h post-burn using the RNeasy PowerSoil Total RNA kits and RNeasy PowerSoil DNA Elution kits (QIAGEN) following manufacturer instructions. Residual DNA contamination was removed from the RNA extracts using DNase Max kits (QIAGEN) following manufacturer instructions. RNA reverse transcription was carried out using Invitrogen SuperScript IV VILO Master Mix (ThermoFisher) following manufacturer instructions. Total RNA and DNA were quantified using a Quant-iT RiboGreen RNA Assay kit (ThermoFisher) and Invitrogen Quant-iT PicoGreen dsDNA Assay kits (ThermoFisher), respectively. For the fast growth and post-fire affinity incubations, DNA was extracted from soil collected at the end of the incubations using the DNeasy PowerLyzer PowerSoil DNA Extraction kit following manufacturer instructions.

Copy DNA and DNA were amplified via triplicate PCR, targeting the 16S rRNA gene v4 region of bacteria and archaea with 515f and 806r primers[74], with barcodes and Illumina sequencing adapters added following ref. [75] (all primers are listed in Supplementary Tables 15 and 16). PCR amplicon triplicates were pooled, purified, normalized and sequenced using 2 × 250 paired-end Illumina MiSeq sequencing at the UW-Madison Biotechnology Center.

## Sequence processing and taxonomic assignments

We quality filtered, trimmed, dereplicated, learned errors, picked OTUs and removed chimaeras using DADA2 (ref. [76]) as implemented in QIIME2 (ref. [77]). Taxonomy was assigned using the naïve Bayes classifier[78] in QIIME2 with the aligned 515f-806r region of the 99% OTUs from the SILVA database (SILVA 138 SSU)[79–81].

## Statistical analyses and trait assignments

We worked primarily in R[82], relying extensively on the R packages phyloseq[83], dplyr[84], ggplot2 (ref. [85]) and vegan[86]. We identified taxa with the traits of interest (survival, fast growth or post-fire environment affinity) from each experiment using the corncob package[87] in R. To identify fire survivors, we estimated the $\log_2$(fold change) in the relative abundance of bacterial DNA from taxa enriched in burned vs unburned soil at 24 h post-burn. To avoid organisms that had DNA present but were actually dead, we excluded taxa with a 16S rRNA:16S rRNA gene ratio less than 1 (ref. [88]). To identify fast growers, we estimated the $\log_2$(fold change) in the relative abundance of bacterial DNA from taxa enriched in burned soil at 5 weeks vs 24 h post-fire. To identify taxa with post-fire environment affinity, we estimated the $\log_2$(fold change) in the relative abundance of bacterial DNA from taxa enriched in burned vs unburned soil following inoculation and a 6-month incubation. For all three traits, we wanted to limit our analysis to samples that definitely experienced ecologically meaningful burn conditions. Thus, we excluded 21 burned soil cores (all moist burns and three dry burns) that did not reach temperatures >50 °C[17] and corresponding control cores, leaving 32 cores with organic horizons and 24 cores with mineral horizons. For all trait identifications, we analysed organic and mineral horizons separately, controlled for pH (to allow for the identification of different responsive taxa across sites with a relatively wide initial range of pHs), excluded statistically significant but small positive responses by excluding taxa with a $\log_2$(fold change) less than one standard deviation below the mean across each experiment and used a false discovery rate cut-off of 0.05 to adjust $P$ values of differentially abundant taxa to account for false positives.

To characterize microbial C mineralization, we were interested in the relative mineralizability of the remaining C in each sample. Using $CO_2$ fluxes on a per-gram total C basis, we fit two-pool exponential decay models for each sample, estimating respiration rate constants and fractional sizes of fast and slow C pools[89] (equation 1),

$$M_t = M_1 e^{(-k_1 t)} + M_2 e^{(-k_2 t)} \tag{1}$$

where $M_t$ is the total C pool, $M_1$ and $M_2$ are the fast and slow C pools, respectively, $k_1$ and $k_2$ are the respiration rate constants for the fast and slow C pools, respectively, and $t$ is time. This model was fit using the nls.lm function in the minpack.lm package[90] in R.

To compare soil properties across horizons and burn treatments, we used ANOVA, Tukey's HSD, and Wilcoxon rank-sum tests and fitted linear models using the stats package[91] in R. To assess predictive factors for whole-community composition, we used PERMANOVA on weighted UniFrac distances[92] calculated from a phylogenetic tree built with FastTree[93] from sequences aligned using the mafft programme[94,95]. To estimate mean predicted 16S rRNA gene copy number, we used the rrnDB RDP Classifier tool (v.2.12) to predict a mean 16S rRNA gene copy number for each OTU[33] and calculated community-weighted mean predicted rRNA gene copy number[22].

## Quantifying post-fire strategies in field data across a burn severity gradient

To determine the importance of each trait in the field, we quantified the total relative abundance of taxa assigned to each of the strategies using our laboratory experiments in a dataset of natural wildfires from the same region[6,32]. Briefly, soils were collected from organic and mineral horizons in the field at 1 and 5 yr post-fire across a range of burn severities including unburned sites, and spanning the same vegetation communities and soil types from which cores were collected for the laboratory experiments. OTUs were identified for the field data using similar methods as for the laboratory experiments (same primers, library preparation, sequencing pipeline and OTU-picking algorithms)[6,32], allowing us to merge the laboratory and field datasets using exact matches for OTUs (that is, amplicon sequence variants). To determine whether there was a significant correlation between burn severity and the total relative abundance of all taxa with a given trait, we fit linear models, including interaction terms between years since the fire (1 or 5) and burn severity index[96]. If the interaction was not significant, we report results from the model without the interaction.

## Reporting summary

Further information on research design is available in the Nature Portfolio Reporting Summary linked to this article.

## Data availability

The sequencing datasets generated during the current study are available in the NCBI SRA under bioproject number PRJNA913093. Field sequencing data are available at PRJNA564811 (2015 data) and PRJNA825513 (2019 data). Non-sequencing data are deposited in the DOE ESS-DIVE repository at https://doi.org/10.15485/1959350. Source data are provided with this paper.

## Code availability

Code for the analyses conducted in the study is available at https://github.com/DanaBJohnson/WoodBuffalo2019.

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

## Acknowledgements

This work was funded by the University of Wisconsin-Madison Fall Research Competition Grant to T.W. and the US Department of Energy (DE-SC0021022) to T.W., both of which supported D.B.J. We thank E. Whitman for assistance in identifying field sites and burn history and for support in field collection; D. K. Thompson for insights on burn characteristics for this region; L. Hasburgh and K. Bourne at the USDA FS Forest Products Laboratory for advice and assistance with burn simulations; J. Morin, S. Irwin and other Wood Buffalo National Park staff for support in conducting this research (Permit WB-2019-31497).

## Author contributions

D.B.J. and T.W. conceptualized the project. D.B.J., J.W., K.M.Y. and T.W. developed the methodology. D.B.J. and T.W. developed software. D.B.J. and T.W. conducted formal analysis. D.B.J., J.W., K.M.Y. and T.W. conducted investigations. D.B.J. wrote the original draft. D.B.J., J.W., K.M.Y. and T.W. reviewed and edited the manuscript. D.B.J. and T.W. performed visualization. K.M.Y. acquired resources and curated the data. T.W. supervised and administered the project and acquired funding.

## Competing interests

The authors declare no competing interests.

## Additional information

**Extended data** is available for this paper at https://doi.org/10.1038/s41559-023-02135-4.

**Correspondence and requests for materials** should be addressed to Thea Whitman.

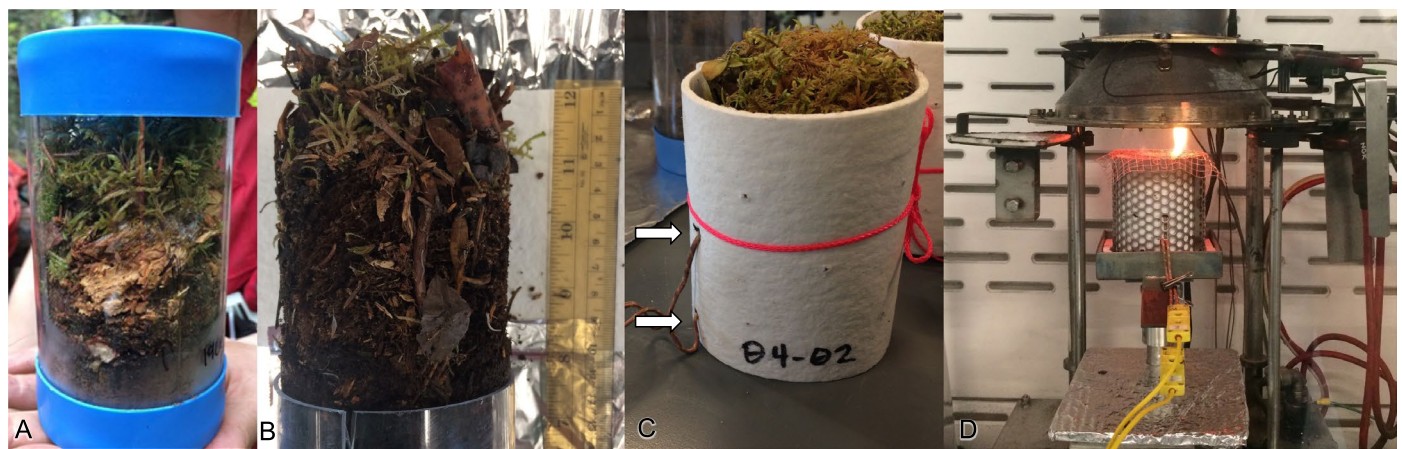

**Extended Data Fig. 1 | Collection and burning of soil cores.** Soil cores (**a**) collected, (**b**) extruded, (**c**) wrapped in ceramic paper in preparation for the burn. White arrows indicate the position of the mid and lower thermocouples within the core. (**d**) Soil cores exposed to heat flux in mass loss calorimeter.

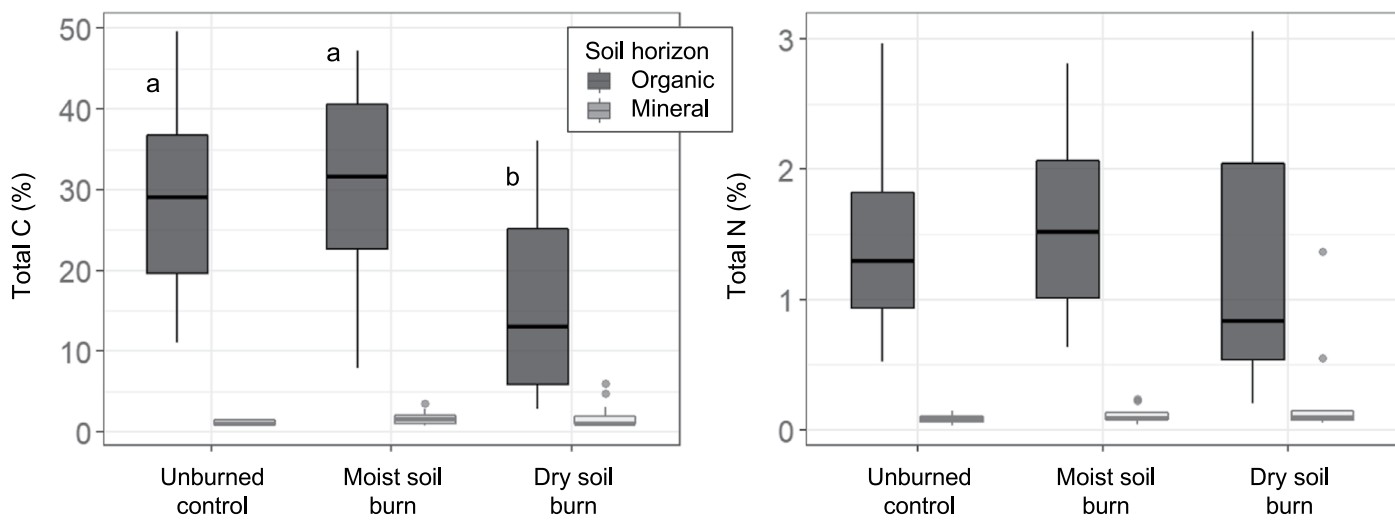

**Extended Data Fig. 2 | Soil total C and total N across the three burn treatments.** C is indicated in the left panel and N in the right panel. Different letters (above each boxplot) represent statistically significant treatment differences based on ANOVA and Tukey's HSD ($n = 18$ for all O horizons, $n = 10$ for unburned mineral horizons, $n = 11$ for moist burn mineral horizons and $n = 12$ for dry burn mineral horizons; $P < 0.01$). For both boxplots, the central horizontal line indicates the median, the upper and lower bounds of the box indicate the inter-quartile range (IQR), the upper and lower whiskers reach the largest or smallest values within a maximum of 1.5 × IQR, and data beyond the whiskers are indicated as individual points.

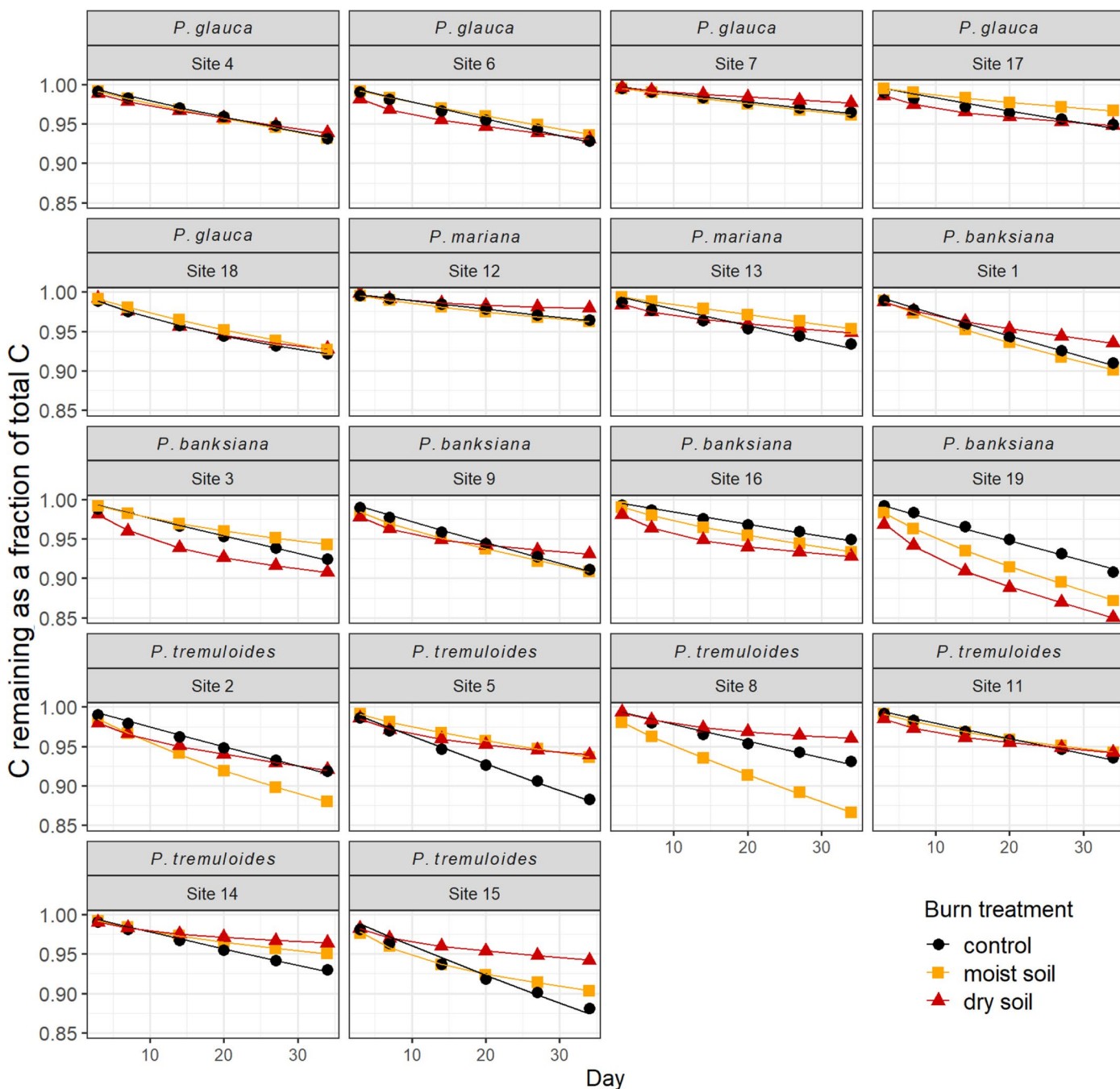

**Extended Data Fig. 3 | Carbon remaining as a fraction of initial total C in burned and unburned soil across all sites.** Fitted lines indicate 2-pool decay models; data are for the 5-week fast-growth incubation.

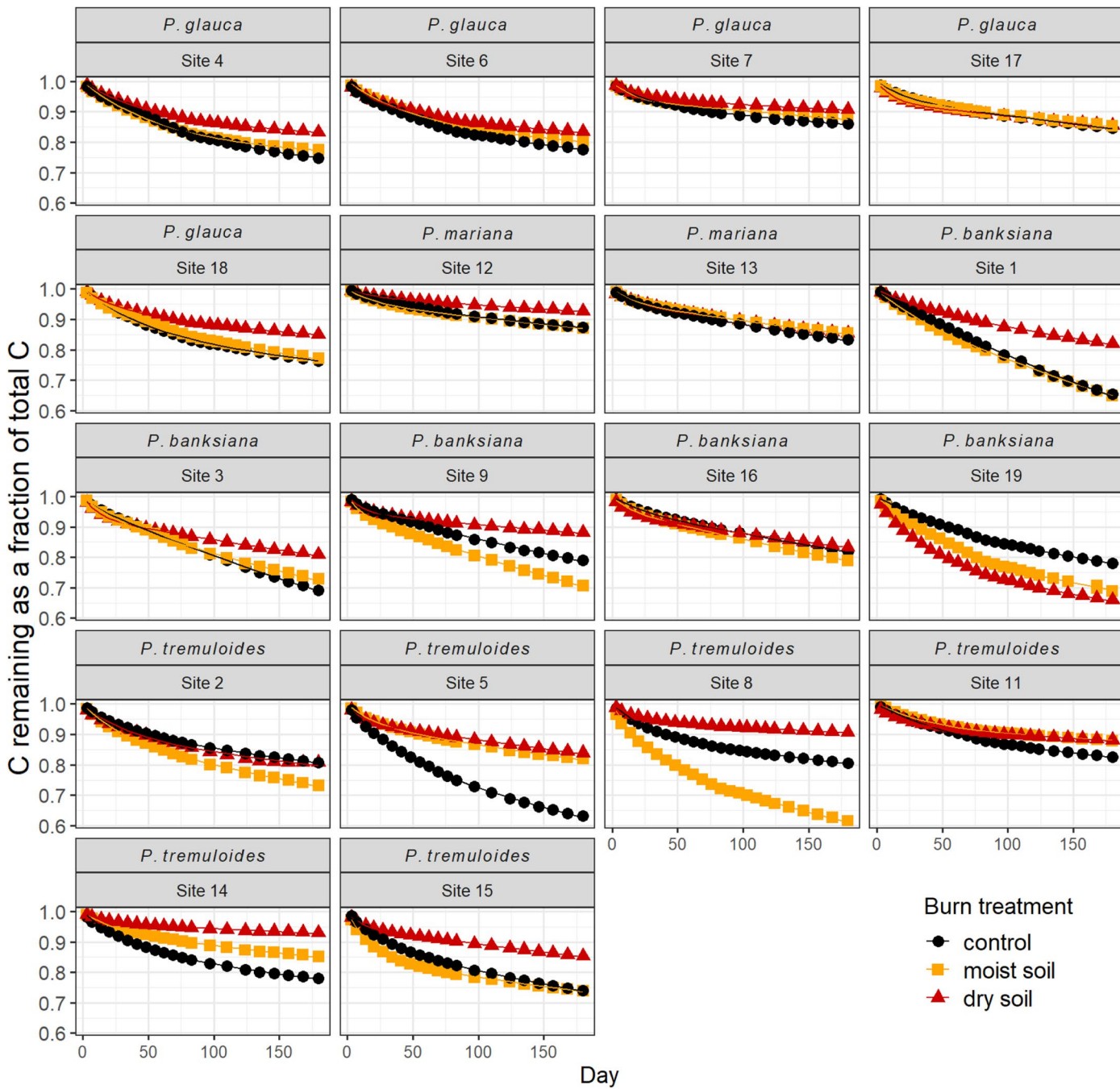

**Extended Data Fig. 4 | Carbon remaining as a fraction of initial total C in burned and unburned soil across all sites.** Fitted lines indicate 2-pool decay models; data are for the 6-month post-fire affinity incubation following autoclaving and inoculation with unburned soil.

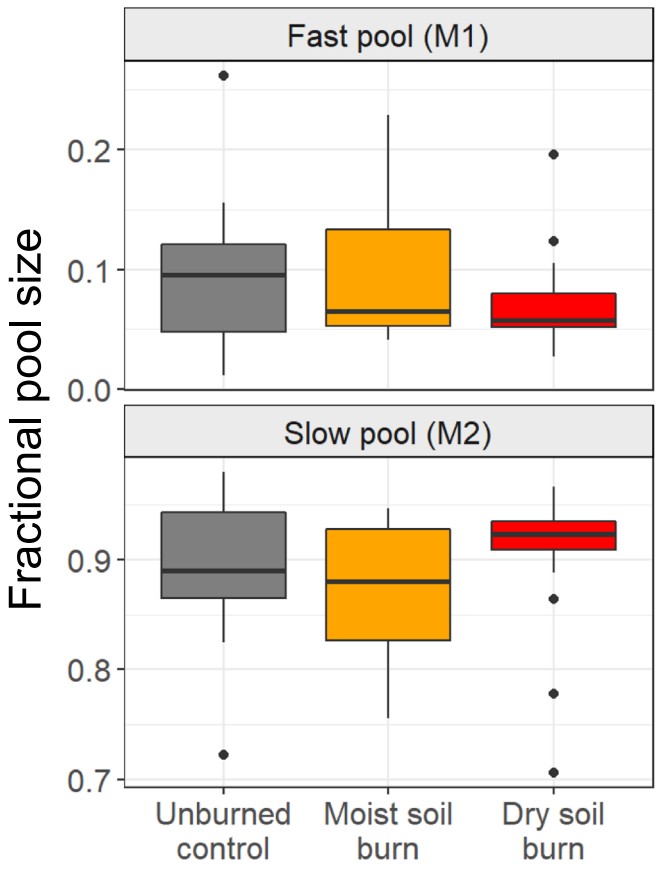

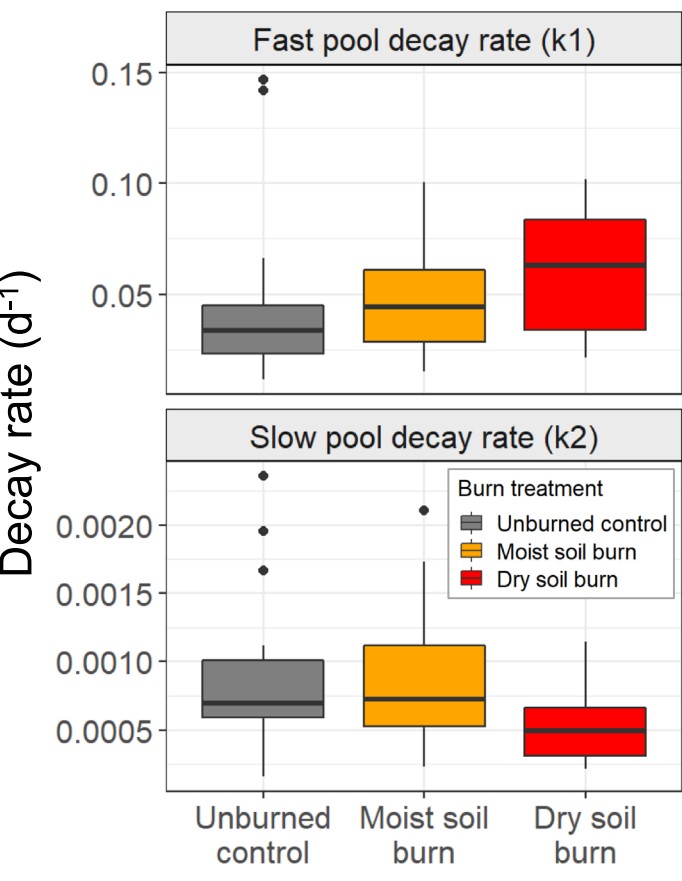

**Extended Data Fig. 5 | Decay model parameters for 6-month post-fire environment affinity incubation.** Coefficients for two-pool exponential decay models fit to soil C respiration data ($R^2$ of models 0.98-1, $P < 0.0001$; $n = 18$ for unburned and moist soil burn cores, and $n = 19$ for dry soil burn cores). Fractional size of the fast ($M_1$) and slow ($M_2$) C pool (left two panels) and decay rate coefficients for the fast ($k_1$) and slow ($k_2$) C pool (right two panels) of the slow

C pool following post-fire environment affinity incubation (autoclaving soil, inoculation with unburned microbial community, and 6-month incubation). For each boxplot, the central horizontal line indicates the median, the upper and lower bounds of the box indicate the inter-quartile range (IQR), the upper and lower whiskers reach the largest or smallest values within a maximum of 1.5 × IQR, and data beyond the whiskers are indicated as individual points.

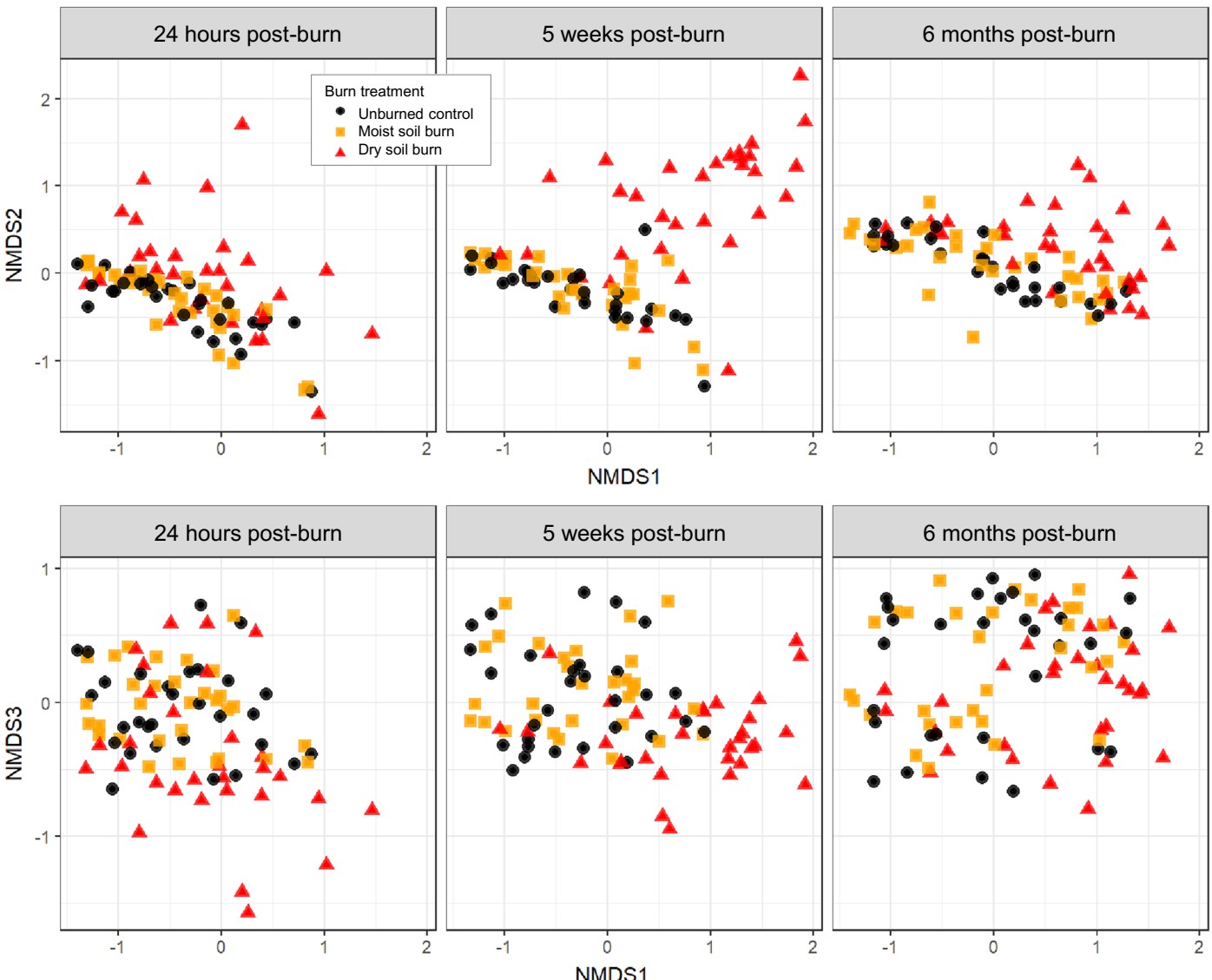

**Extended Data Fig. 6 | Community composition of laboratory soil samples.** NMDS ordination on weighted UniFrac distance of genomic DNA 16S rRNA genes across all experiments ($k = 3$, stress=0.12, $n = 93$ soil horizons). The top plot indicates axes 1 and 2; the bottom plot indicates axes 1 and 3. One ordination was performed and is faceted by the three experimental timepoints – 24 hours, 5 weeks, and 6 months (following autoclaving and inoculation with unburned soil). For all communities across the three laboratory experiments, all tested factors (dominant vegetation, soil horizon, pH, burn treatment, and incubation treatment) were significant predictors of community composition in combined models (PERMANOVA, p = 0.001 for all factors). Soil pH provided the most explanatory power ($R^2 = 0.07$) for community composition followed by experimental treatment ($R^2 = 0.05$).

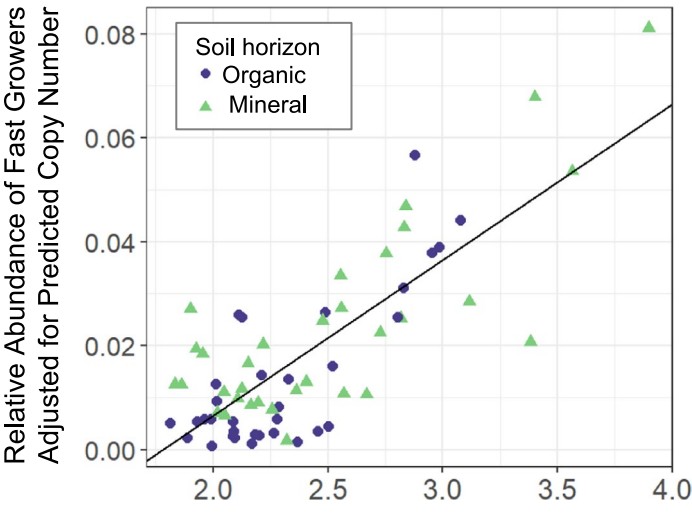

**Extended Data Fig. 7 | Weighted mean predicted 16S rRNA gene copy number is correlated with relative abundance of lab-identified fast growers in field data.** Weighted mean predicted 16S rRNA gene copy number vs relative abundance of lab-identified fast growers adjusted for mean predicted 16S rRNA gene copy number in field data one year post-fire. Line indicates linear fit ($n = 66$ field soil samples; $P = 2 \times 10^{-15}$, $R^2_{adj} = 0.62$, $y = -0.05 + 0.03x$).

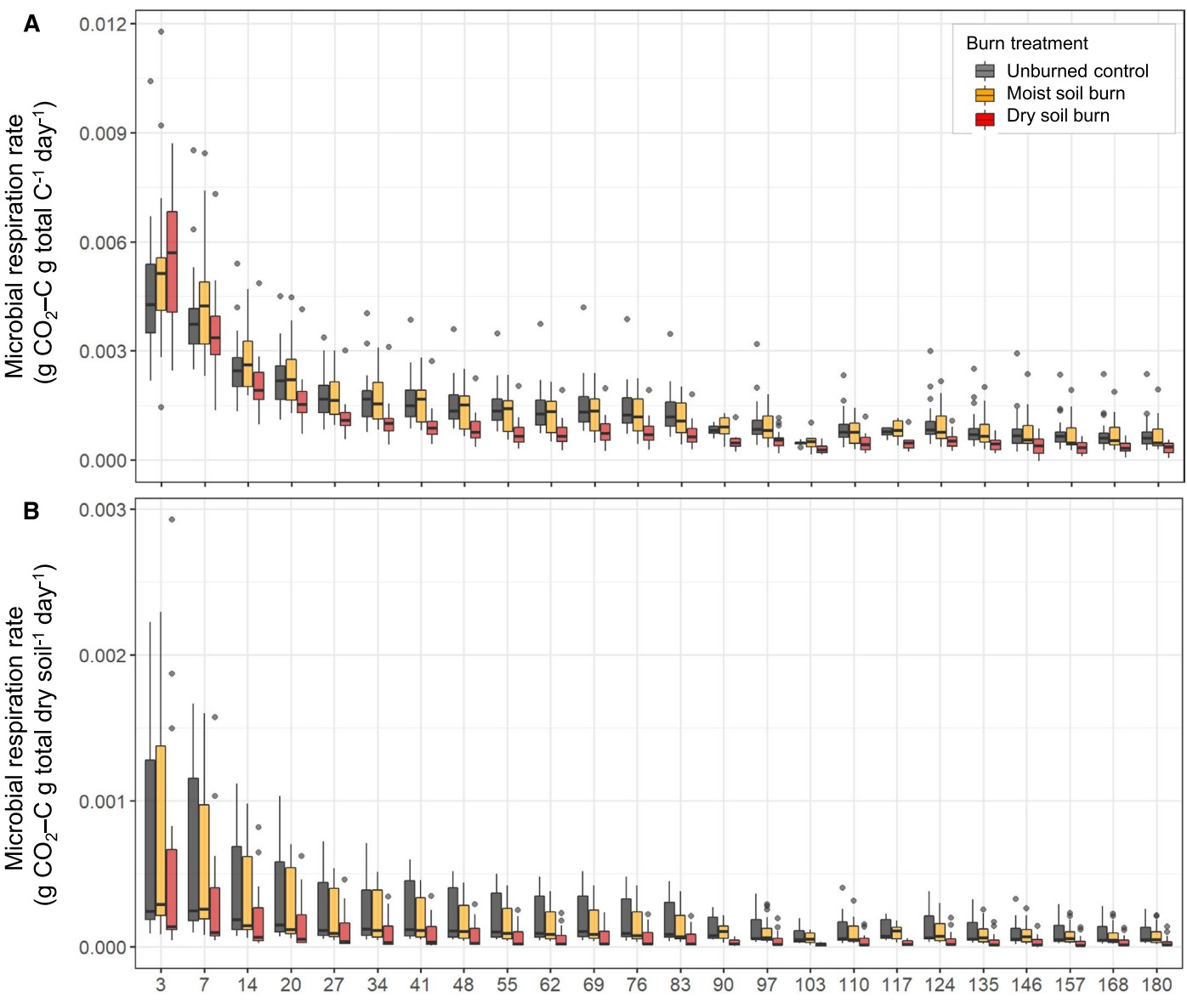

**Extended Data Fig. 8 | Respiration rates for lab-incubated soils.** Respiration rate per (**a**) gram initial total C and (**b**) gram dry soil for unburned (*n* = 18 soil cores), moist soil burn (*n* = 19 soil cores), and dry soil burn (*n* = 19 soil cores) over the course of the 6-month incubation, following autoclaving and inoculation with unburned soil. For each boxplot, the central horizontal line indicates the median, the upper and lower bounds of the box indicate the inter-quartile range (IQR), the upper and lower whiskers reach the largest or smallest values within a maximum of 1.5 × IQR, and data beyond the whiskers are indicated as individual points.

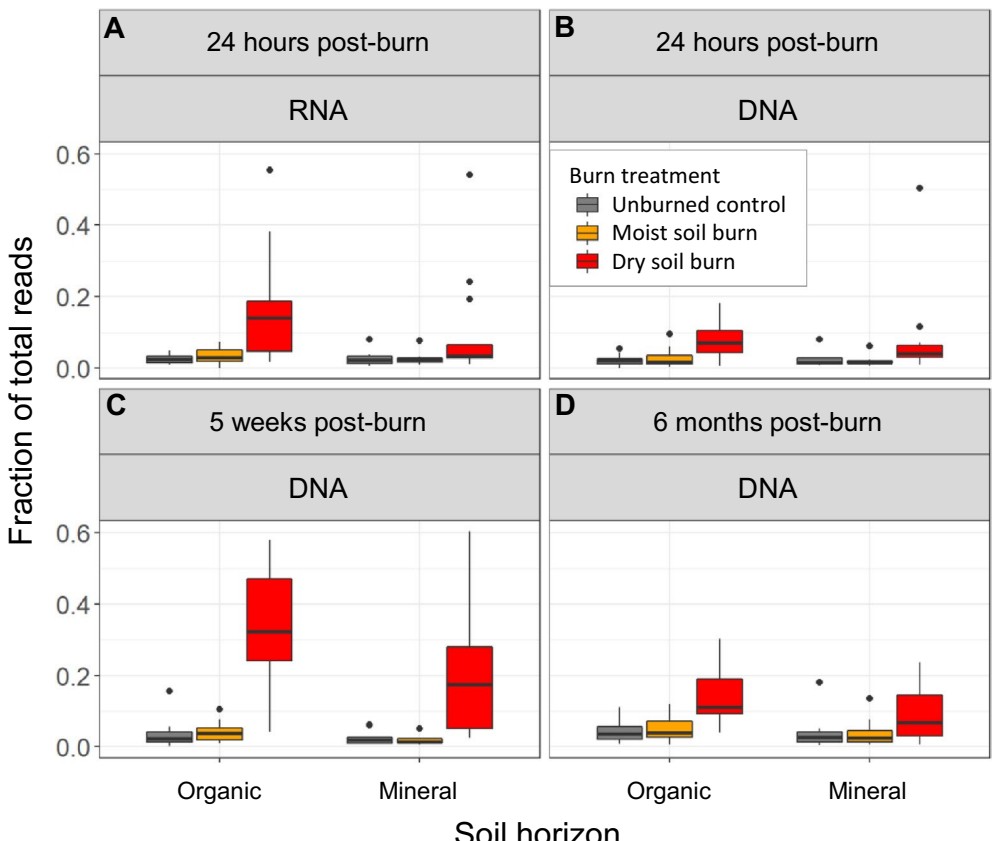

**Extended Data Fig. 9 | The fraction of total reads identified as taxa with any of the three fire-adaptive strategies.** (**a**) rRNA ($n$ = 19 for unburned O horizons, $n$ = 19 for moist burn O horizons, $n$ = 18 for dry burn O horizons, $n$ = 11 for unburned mineral horizons, $n$ = 12 for moist burn mineral horizons and $n$ = 13 for dry burn mineral horizons) and (**b**) DNA samples immediately post-fire ($n$ = 19 for unburned O horizons, $n$ = 18 for moist burn O horizons, $n$ = 18 for dry burn O horizons, $n$ = 11 for unburned mineral horizons, $n$ = 12 for moist burn mineral horizons and $n$ = 13 for dry burn mineral horizons), (**c**) 5-week fast growth incubation ($n$ = 19 for O horizons of all treatments, $n$ = 11 for unburned mineral horizons, $n$ = 12 for moist burn mineral horizons and $n$ = 12 for dry burn mineral horizons), and (**d**) 6-month post-fire environmental affinity incubation ($n$ = 19 for O horizons of all treatments, $n$ = 11 for unburned mineral horizons, $n$ = 12 for moist burn mineral horizons and $n$ = 13 for dry burn mineral horizons) following autoclaving and inoculation with unburned soil. The central horizontal line indicates the median, the upper and lower bounds of the box indicate the inter-quartile range (IQR), the upper and lower whiskers reach the largest or smallest values within a maximum of 1.5 × IQR, and data beyond the whiskers are indicated as individual points.

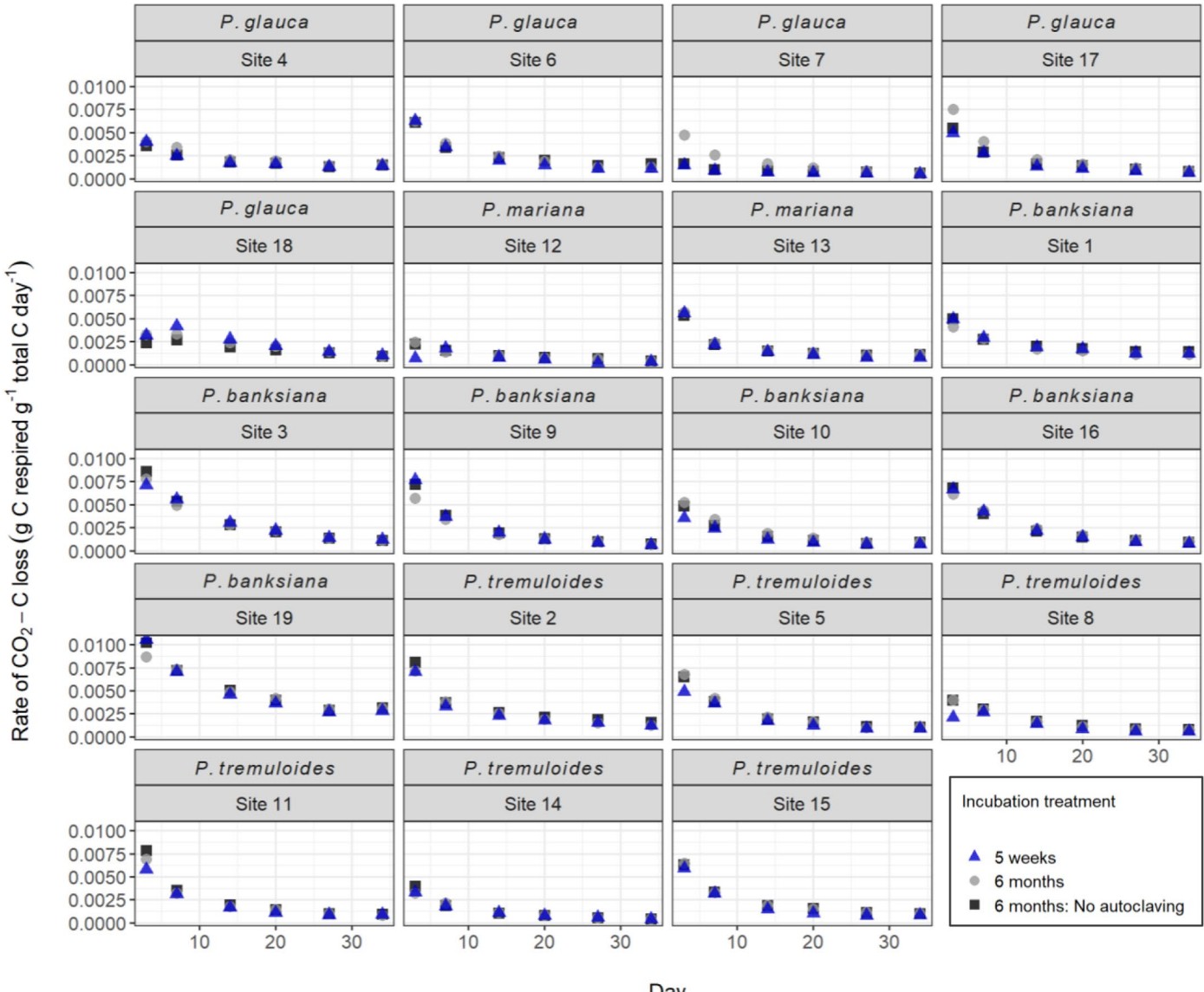

**Extended Data Fig. 10 | Respiration rates are similar with or without inoculation.** Comparison of respiration rate (per initial gram C) for dry burn soil over the first 5 weeks of the 5-week fast growth incubation, 6-month post-fire environment affinity incubation including autoclaving and inoculation with unburned soil microbial community, and 6-month incubation including inoculation but excluding autoclaving.

# Reporting Summary

## Statistics

For all statistical analyses, confirm that the following items are present in the figure legend, table legend, main text, or Methods section.

| n/a | Confirmed | |
|---|---|---|
| ☐ | ☒ | The exact sample size (*n*) for each experimental group/condition, given as a discrete number and unit of measurement |
| ☐ | ☒ | A statement on whether measurements were taken from distinct samples or whether the same sample was measured repeatedly |
| ☐ | ☒ | The statistical test(s) used AND whether they are one- or two-sided <br> *Only common tests should be described solely by name; describe more complex techniques in the Methods section.* |
| ☐ | ☒ | A description of all covariates tested |
| ☐ | ☒ | A description of any assumptions or corrections, such as tests of normality and adjustment for multiple comparisons |
| ☐ | ☒ | A full description of the statistical parameters including central tendency (e.g. means) or other basic estimates (e.g. regression coefficient) AND variation (e.g. standard deviation) or associated estimates of uncertainty (e.g. confidence intervals) |
| ☐ | ☒ | For null hypothesis testing, the test statistic (e.g. *F*, *t*, *r*) with confidence intervals, effect sizes, degrees of freedom and *P* value noted <br> *Give P values as exact values whenever suitable.* |
| ☒ | ☐ | For Bayesian analysis, information on the choice of priors and Markov chain Monte Carlo settings |
| ☒ | ☐ | For hierarchical and complex designs, identification of the appropriate level for tests and full reporting of outcomes |
| ☒ | ☐ | Estimates of effect sizes (e.g. Cohen's *d*, Pearson's *r*), indicating how they were calculated |

*Our web collection on statistics for biologists contains articles on many of the points above.*

## Software and code

Policy information about availability of computer code

| | |
|---|---|
| Data collection | All data were collected in the lab, not using software or code. |
| Data analysis | All analyses in this paper were conducted in R, as detailed in the methods section. Code for the analyses conducted in the current study are available at https://github.com/DanaBJohnson/WoodBuffalo2019. |

For manuscripts utilizing custom algorithms or software that are central to the research but not yet described in published literature, software must be made available to editors and reviewers. We strongly encourage code deposition in a community repository (e.g. GitHub). See the Nature Portfolio guidelines for submitting code & software for further information.

## Data

Policy information about availability of data

All manuscripts must include a data availability statement. This statement should provide the following information, where applicable:
- Accession codes, unique identifiers, or web links for publicly available datasets
- A description of any restrictions on data availability
- For clinical datasets or third party data, please ensure that the statement adheres to our policy

The sequencing datasets generated during the current study are available in the NCBI SRA under bioproject number PRJNA913093. Field sequencing data are available at PRJNA564811 (2015 data) and PRJNA825513 (2019 data, available 31 December 2022). Non-sequencing data are deposited in the DOE ESS-DIVE repository under: https://data.ess-dive.lbl.gov/datasets/ess-dive-e00cfbe3d89f2c1-20220421T224630867

# Human research participants

Policy information about studies involving human research participants and Sex and Gender in Research.

| | |
|---|---|
| Reporting on sex and gender | NA |
| Population characteristics | NA |
| Recruitment | NA |
| Ethics oversight | NA |

Note that full information on the approval of the study protocol must also be provided in the manuscript.

# Field-specific reporting

Please select the one below that is the best fit for your research. If you are not sure, read the appropriate sections before making your selection.

☐ Life sciences  ☐ Behavioural & social sciences  ☒ Ecological, evolutionary & environmental sciences

For a reference copy of the document with all sections, see nature.com/documents/nr-reporting-summary-flat.pdf

# Ecological, evolutionary & environmental sciences study design

All studies must disclose on these points even when the disclosure is negative.

| | |
|---|---|
| Study description | Our overarching goal is to develop a fire ecology framework for soil microbes. Specifically, we aimed to identify bacteria with fire-related traits (fire survival, fast growth, and affinity for the post-fire environment), and (1) assess how the abundance of taxa with these different traits vary one and five years post-fire, and with increasing burn severity (2) determine whether there were trade-offs between these traits, and (3) determine how fire-induced changes to soil microbes and soil properties affect post-fire C cycling.<br><br>To this end, we collected soil cores from 19 sites. For each site, 3 fire treatments were applied to an individual core - control, moist soil burn, dry soil burn (i.e., N=19 for each burn treatment). Each site x fire treatment was then included in each of three experiments designed to identify fire-responsive traits (survival, fast growth, and affinity for the post-fire environment). Subsequent analyses were performed on organic and mineral horizons separately. Since some sites had no mineral horizon within the sampled depth (particularly in the Picea sp.-dominated sites), N=19 for organic horizons and N=12 for mineral horizons. |
| Research sample | The samples are soils collected from 19 sites across Wood Buffalo National Park in the Northwest Territories and Alberta, Canada. Samples were collected from 7 Picea sp.-dominated sites, 6 Pinus banksiana-dominated sites, and 6 Populus tremuloides-dominated sites that had not burned in the last 30 years. These sites were chosen using stratified random sampling using the Canadian National Fire Database, dominant vegetation from the Canadian National Forest Inventory, and soil type from FAO soil survey data, with the goal of representing the dominant vegetation and soil types of the southern half of the boreal and taiga plains ecoregions of northwestern Canada.<br><br>The study draws on bacterial community sequencing data from T Whitman, J Woolet, M Sikora, DB Johnson, E Whitman. 2022. Resilience in soil bacterial communities of the boreal forest from one to five years after wildfire across a severity gradient. Soil Biology and Biochemistry 172, 108755, which are available in the NCBI SRA at PRJNA564811 (2015 data) and PRJNA825513 . |
| Sampling strategy | Sites that had not burned in the previous 30 years or more were selected using stratified random sampling using the Canadian National Fire Database, dominant vegetation from the Canadian National Forest Inventory, and soil type from FAO soil survey data. Sites were located between 0.1 and 1 km from roads and > 0.5 km from other sampling sites. We used a Garmin GPSMAP 64 GPS finder to reach each designated location. Upon arriving, we confirmed the dominant tree species and recorded slope and aspect. Samples were collected across a 2 x 2-meter grid. A collapsible PVC pipe square was used to map out the sampling grid. After sampling most field sites, we produced a second selection of random points in a more limited region with field-validated tree species dominance. This second random sample was designed to address identified gaps in species dominance in the initial sample that were the result of errors and limitations in the map products used, resulting in a total of 19 sites, with 6-7 sites under each dominant vegetation type. At each site, ten soil cores (15.24 cm x 7.62 cm dia.) were collected using a soil core sampler with clear plastic core liners and plastic end caps (Product IDs 405.09 and 418.09; AMS, American Falls, ID, USA) every 1 m within (and two at the center of) a 2 m x 2 m grid.<br><br>Sample size and sampling strategy were chosen to maximize representation of the region, using the stratified random sampling approach described above. This sample number was sufficient to achieve our goal for this sampling design, which was to identify bacterial taxa with fire-responsive traits in this region: in the lab-burned dry cores, these taxa represented a mean of 32% of the total reads. With a higher sample size, we might have been able to conclusively identify lower-abundance or rarer taxa that have these traits. However, given these taxa tend to be lower-abundance and rarer across samples in the first place, we expect that adding them to our list of trait-identified taxa would have relatively small effects on the main trends illustrated in Figures 4-6, and also means our approach is generally conservative. For soil properties, we were able to detect significant changes in C, C:N, and pH, using standard statistical tests (ANOVA, Wilcoxon signed rank test), indicating our sample size was sufficiently large to assess these changes. Where |

| | |
|---|---|
| | we did not detect significant differences (e.g., total N%), the effect sizes were generally relatively small, and perhaps of limited ecological relevance. They also are explainable - for example, during combustion of OM, C losses begin around 100 °C, whereas N volatilization begins around 200 °C, so it makes sense that we detected significant differences in %C but not %N. |
| Data collection | Soil texture was determined using a physical analysis hydrometer at the UW-Madison Soil and Forage Lab. Soil pH was measured by DBJ as detailed in Supplementary. Soil subsamples were prepped for total C and N analysis by DBJ and analyzed as detailed in Methods. Soil respiration was measured by DBJ as detailed in Supplementary. RNA and DNA concentrations were measured by DBJ as detailed in Methods and Supplementary. RNA and DNA extraction was carried out by DBJ and sequencing was done by the UW-Madison Biotechnology Center as described in Methods and Supplementary. |
| Timing and spatial scale | We collected 10 soil cores each from 19 sites within Wood Buffalo National Park in the Northwest Territories and Alberta, Canada between June 13 and 17, 2019. One to seven sites per day were sampled and each site was only sampled once in an effort to collect all samples within a relatively short time period to avoid confounding effects of seasonality. Exact latitude and longitude of each site are given in Supplemental Table S1. |
| Data exclusions | C and N values from site 10 control, moist burn soil, and dry burn soil were excluded for all relevant analyses due to an error at time of measurement. We excluded one cDNA and two gDNA samples from further analysis due to low 16S reads per sample (<1000). |
| Reproducibility | We have not yet attempted to reproduce the experiment in its entirety, from core collection to final analyses. However, we intentionally designed our approach to be as replicable as possible. Specific elements to improve replicability include the following: Allowing cores to air dry to moisture contents typical of drought in the region of study before imposing moisture and burn treatments means these treatments could be repeated with high replicability. Using a cone calorimeter to deliver specific heat fluxes, representative of a crown fire in this region, means the burn simulations are also highly replicable. Subsequent experiments and analyses draw on standard methods and protocols wherever possible, ensuring replicability is likely. Finally, all our data and analytical code are public, allowing for our analyses to be reproduced. |
| Randomization | Samples were not allocated into groups - every treatment was applied to every sample. |
| Blinding | During data collection and analysis, all samples were given a numeric identity that did not indicate their related treatment. We did not attempt to mask soil horizon identity because it is visually distinctive. |

Did the study involve field work?  ☒ Yes  ☐ No

## Field work, collection and transport

| | |
|---|---|
| Field conditions | Temperature during sampling ranged from 10-30 °C and total precipitation over the 5 day sampling period was 8.4 mm. |
| Location | Exact sampling locations and site properties are given in detail in Supplemental Table S1. |
| Access & import/export | All samples were collected within the Wood Buffalo National Park under permit # WB-2019-31497 |
| Disturbance | Fieldwork was designed to generally be low-impact, restricted to hiking by foot into the sites and collecting small soil cores, spaced at least 1 m from each other. We worked with the parks staff to ensure sensitive areas were not sampled. |

# Reporting for specific materials, systems and methods

We require information from authors about some types of materials, experimental systems and methods used in many studies. Here, indicate whether each material, system or method listed is relevant to your study. If you are not sure if a list item applies to your research, read the appropriate section before selecting a response.

### Materials & experimental systems

| n/a | Involved in the study |
|---|---|
| ☒ ☐ | Antibodies |
| ☒ ☐ | Eukaryotic cell lines |
| ☒ ☐ | Palaeontology and archaeology |
| ☒ ☐ | Animals and other organisms |
| ☒ ☐ | Clinical data |
| ☒ ☐ | Dual use research of concern |

### Methods

| n/a | Involved in the study |
|---|---|
| ☒ ☐ | ChIP-seq |
| ☒ ☐ | Flow cytometry |
| ☒ ☐ | MRI-based neuroimaging |

