## [Peer Review File · Nature Ecology & Evolution]

Peer Review Information

Journal: Nature Ecology & Evolution

Manuscript Title: Experimentally-determined traits shape bacterial community composition one and five years following wildfire

Corresponding author name(s): Thea Whitman

Editorial Notes:

Reviewer Comments & Decisions:

Decision Letter, initial version:

19th January 2023

Dear Dr Whitman,

Your manuscript entitled "Experimentally-determined traits shape bacterial community composition one and five years following wildfire" has now been seen by three reviewers, whose comments are copied below. The reviewers have raised a number of concerns which we would like to see addressed in a revised manuscript before we can reach a final decision regarding publication in Nature Ecology & Evolution. In particular, the comments from Reviewers 1 and 3 indicate that the conceptual basis of the work needs to be reconsidered, and several methodological aspects need more detail and justification.

We therefore invite you to revise your manuscript taking into account all reviewer and editor comments. Please highlight all changes in the manuscript text file.

* If you have not done so already please begin to revise your manuscript so that it conforms to our Article format instructions at <http://www.nature.com/natecolevol/info/final-submission>. Refer also to any guidelines provided in this letter.

2[REDACTED]

Nature Ecology & Evolution is committed to improving transparency in authorship. As part of our efforts in this direction, we are now requesting that all authors identified as 'corresponding author' on published papers create and link their Open Researcher and Contributor Identifier (ORCID) with their account on the Manuscript Tracking System (MTS), prior to acceptance. ORCID helps the scientific community achieve unambiguous attribution of all scholarly contributions. You can create and link your ORCID from the home page of the MTS by clicking on 'Modify my Springer Nature account'. For more information please visit <http://www.springernature.com/orcid>.

[REDACTED]

Reviewer expertise:

Reviewer #1: Microbial responses to disturbance

Reviewer #2: Fire ecology, soil carbon

Reviewer #3: Microbial traits and functions

Reviewers' comments:

Reviewer #1 (Remarks to the Author):

Review of Johnson et al. Nat Eco Evol

The work combines field and laboratory experiments to assign bacterial traits hypothesized to relevant

2for carbon fluxes and for understanding the general microbial community ecology of forest ecosystems that experience fire regimes. Strengths of the work include the complementary lab/field approach as well as the measurement of soil factors to relate to the microbiological patterns observed. I appreciated the conceptual framework with new hypotheses to suggest future directions, which is provided at the end of the work.

The three non-mutually exclusive traits predicted to have importance (fire survival, fast growth within 5 weeks of burn, fitness post-burn at 6 months) were not observed in many taxa generally, and were not found to be highly explanatory of community outcomes or carbon fluxes. The authors concede this, and then suggest that plant community assembly and dormancy may be more important. These two alternative hypotheses are not addressed by data presented in the current work. Thus, despite the interesting and novel experimental design, there are not strong conclusions to be drawn because the major trait hypotheses are not supported. Also, there are too few taxa with each trait/combinations of traits detected to make strong predictions about trade-offs or relative strengths among the traits.

There are a lot of different types of data included in this work, and as a reader I found it difficult to keep track of everything, having to often move between different sections and between main and supplemental. The manuscript could be improved with a clearer focus and by omitting results that are not directly contributing to the main outcome.

Is there a possibility that how the traits were defined – based on differential DNA-based taxa abundances between appropriately controlled lab experimental conditions and then determining their collective, proportional contribution to the total community with weighted copy numbers– is somehow missing the mark for capturing them in the field? There are a few reasons why this may be.

One reason is that there are notably different scales of the lab v. field observations. In the lab, the taxon traits defined at differential abundances at 24 hours, 5 weeks, and 6 months, but in the field there is the 1 v. 5 years post-burn. It could be that many taxa with these traits are not observed because both field time points extend beyond the lab experiments, and so the authors are looking for persistence of taxa with these traits long term without understanding if they were detected in the field at the same time scales as the lab experiments, initially. The field is long-term assembly (press) rather than short term response (pulse). The authors introduce this in the beginning of the work, but these themes are not well integrated into the experimental design. In other words, given what is known about turnover in terrestrial microbial communities, potential historical contingency, and long-term assembly: why would we expect a taxon with a trait that supports fitness at 24 hours, 5 weeks, or 6 months after fire to be similarly fit and active, due to that same trait, at 1 and 5 years after a burn? Of the three traits examined, the only one that may be relevant at 1 and 5 years is “affinity”, but this is an extrapolation beyond the lab experimental design.

Another reason is that the weighted copy numbers may not well-capture the nuances of changes in relative contributions (for the fast growth dataset). The literature is divided on this – some researchers urge that copy number corrections are not legitimate because of database/knowledge gaps. But when applied to construct a community weighted average, overarching bias may soften, but then patterns at the taxon level (or summed taxon level across a minor subset of the community, as here) may not emerge. The authors have a handful of taxa that exhibit each trait in the field, and they

3could attempt to validate their relative contributions with a few qPCRs with specifically designed primers to determine whether this may be an issue for the current study. I am not sure if it is worth it, or if it would be better to just use relative abundances instead of applying weights. The authors could compare outcomes when using relative abundances versus weighted – if the general outcomes are the same, then it doesn't matter. If they are different, then the ultimate choice must be motivated and rationalized.

There are a lot of interesting observations in this work for the physico-chemical properties of soil after burn and given moist/dry conditions, burn severity and different soil horizons. In some cases, the authors show that these environmental properties are deterministic/explanatory of community outcomes. There is a sense of disconnect between the environmental response and biological one that distracts from the main framing about traits but at the same time draws interest, because there are statistically supported results to discuss (whereas with the traits there are fewer results of interest...). I appreciate the challenge that exists in prioritizing and integrating all these aspects into a coherently framed piece.

The work has omitted reporting of who were the taxa that were determined to have the traits, i.e. the community structure, or general discussion of the composition/phylogenotypes that were observed in this system. This should be provided, at least in supplemental. A simple bar chart by treatment/condition? Also, richness/number of observed taxa and any differences between treatments/conditions? These are basic information that are expected with microbiome analyses because they help to understand the ecology of the communities and interpret their patterns.

Specific comments

The gray scale figures do not well visually distinguish the treatments of interest, especially in e.g. Figure 2A. If grayscale must be used, suggest at least having open and closed symbols included to help to distinguish the treatments.

L91 PyOM is introduced, leading readers to think that it will be important for the study but it not returned to with data/results. pH, however, is measured. Perhaps focus the introduction only on those pieces of information that provide context for the study. Same with L 498 in discussion- this could be streamlined a bit because PyOM was not directly measured and so speculative.

It is not clear how the RNA extracts were used in this experiment to assess fire survival. They were reverse transcribed and sequenced, but then what? Did the authors apply a ratio threshold of rRNA: rDNA to determine activity? Did the authors determine differential relative abundance by rRNA (cDNA) alone? From Lines 215-216, it seems that this may be the case. If so, this would be a fundamental flaw for interpretation, as sequence counts of rRNA without reference to rDNA are not interpretable as proxies for relative abundance of the organisms. This is due to differences in gene regulation, transcription, and rDNA copy numbers across different taxa, as discussed in the literature (e.g. Blazewicz 2013 ISMEJ). I checked the github repo to try to determine how the rRNA and rDNA sequences were used, and could not deduce. The methods need to be expanded to include this information.

Line 222 If the cores that did not reach >50 were omitted, how many "moist" cores were left?

4Generally, sample sizes are not apparent in the text or reported with the figure legends. It may be useful to have a table that has all the sample sizes per experiment/condition for reference. It also would be good to plot each discrete point on the box plots (or, alternatively, use violin plots with points overlay to convey each distribution quickly). I believe that this journal requires sample sizes to be reported in the figures or figure legends.

L223 re: trade-offs. Only taxa with more than 1 trait can be assessed for trade-offs, but this is conceptually difficult. It could be that trade-offs are complete, but more likely, because these traits are not mutually exclusive in their underlying genes/mechanisms/regulations, that trade-offs are not biologically possible. The authors mention this possibility, but a bigger question is whether it makes sense to assume/assess trade-offs for traits that are known to be not mutually exclusive?

Line 300. The "good correspondence" could be statistically supported? Could the authors report the % overlap in community composition between the field and lab (and number of taxa not detected in the lab that were detected in the field).

Line 304 p-value

Figure 3. It is hard to interpret these figures. Perhaps the lab and field samples could be separated out into their own ordinations so that the differences between the treatments can be better viewed. I assume that the reason the organic v. mineral are shown separately are because these have highest explanatory value in the community patterns? If not, please motivate.

L425. Who were the other fire survivors?

L 454. "Despite this, microbial communities can take longer than five years to recover post-fire...". The authors could quantify the degree of recovery (or similar metric of resilience) for their field dataset.

L533 Community size (number of cells/individuals) was not directly measured? DNA concentration was, and I agree that this may proxy size/biomass, but not precisely.

Given the difference in microbial nucleic acid concentration able to be extracted from different conditions (fig 4A), the community was changing in its total biomass and probably the size. But, because of this, the fraction of total reads/total read abundances that are attributed to each fire-adaptive trait (e.g, Fig S10) are challenging to think about. There could be the same number of taxa in the different conditions that produce different fractions of reads if the community size has changed.... This is not a unique issue with this study but rather a common across all sequencing studies that use relativized abundances. However, because the community biomass/size is expected to change drastically given the disturbance, it is highly relevant here for interpretation. Could these charts also be provided with a y-axis standardized to the total DNA concentration or perhaps qPCR of housekeeping or even 16S genes to enable comparisons given the differences in community biomass/sizes across the treatments? Or perhaps reporting richness will also help to understand how the number of taxa observed varies given the conditions.

Perhaps I missed it, but are the statistics reported that show whether there are differences in

5community structure across the 24 h, 5-week, and 6-month communities in the lab? If not, is there = some discussion to add about how communities can be overall unchanged but particular populations responsive to the treatments, and attributed due to the proposed traits?

I'm not sure that Figure 7 and associated analyses are conclusive for assessing "relative strength" of traits. An improved visualization may be some version of a ternary plot, with each trait serving as the angles of the triangle? But, as far as the relative strengths, this analysis could be omitted given that there is no clear conclusion to glean from it, and omitting it may serve to further focus the manuscript. Rather than strengths/trade-offs (which seem difficult to assess given the limited number of taxa that were observed to have more than 2 traits), I was curious to know which taxa belonged to each trait and trait combination, and whether anything biologically or ecologically is known about them from the literature that may support their inclusion into those assignments.

The rich dataset presented is a bit under-analyzed as far as relating the community assembly patterns to ecological outcomes of both community and carbon stability, resistance, resilience for each treatment (and the potential contribution of the measured traits/taxa to those properties). The work is largely descriptive despite that nice contrasts/comparisons could be quantified for recovery parameters.

Data Availability

The SUB11346170 project was not found on SRA

PRJNA564811 and PRJNA825513 were found and accessible on the SRA

The link to the ESS-DIVE data repository did not work and returned an error.

The GitHub repository is available, and it would strongly benefit from a readme file to orient users to reproduce the analysis. The input and output files for each R script are not available, and so I could not attempt to reproduce the analyses. Importantly, the OTU tables used for analyses could be provided.

Reviewer #2 (Remarks to the Author):

The manuscript entitled "Experimentally-determined traits shape bacterial community composition one- and five-years following a wildfire" combined field and laboratory experiments to examine the ways soil bacterial communities' structure and function change over time following fire events in boreal forests. More specifically, the authors tested microbial survival, growth, and affinity, here considered to represent traits of fire response strategies, in fire-affected soils. The study was well thought and designed, and the manuscript is very well written. I was thought it particularly interesting to see a study of fire impacts that integrates functional ecology and biogeochemistry, with a nice use of field and laboratory studies to test hypotheses. I was particularly excited about the experimental designs to test the consequences of changes in microbial community composition to ecosystem function (here represented as C losses as CO₂ fluxes). I was left wondering, however, whether the title of the paper would be more impactful if it also reflected the results on functional resilience (i.e., CO₂ flux or C mineralization). The study was conducted using soils collected from Boreal forests and we know that fire regimes differ across ecosystem types. I had the impression that, in the discussion

6section, the idea that the results could be more representative of Boreal forests was a bit lost. This is not, however, to discourage the authors to place some of their discussions in the context of fires in general and terrestrial ecosystems. But I thought that placing the discussion of the results in the context of boreal forests (fuels /fire frequency/etc.) prior to discussing what the results means at the ecosystem-level in general, would be more appropriate. Some of the findings were that fire survival was important in structuring soil bacterial communities, although likely under specific/narrow fire conditions, and that the role of (fast) growth in structuring soil microbial communities was stronger at the first-year post-fire than five years following the fire. Also, microbial community size and composition did not alter soil CO₂ fluxes following fires. Soil conditions were considered in lines 479-486, for example, but I do think that both fire and soil conditions are driving these trends. Regarding statements in line 521- 524, I agree that is very hard to disregard the importance of these interactions belowground during ecosystem recovery trajectory. Boreal forest soils are considered a long reservoir of carbon. Fire could affect this reservoir by affecting the soil environment where microbes regulate the cycling of carbon. But we still need to better understand under what conditions and why fires affect release of this stored carbon to the atmosphere in the future. The paper by Johnson et al. contributes to knowledge of how microbes are affected by fires, shedding light to the ecological mechanisms that are more likely to prevail and affect carbon storage during the recovery trajectory following a fire.

Reviewer #3 (Remarks to the Author):

This study thoroughly sheds light on the ecology of post-fire bacterial communities through the lens of adaptive traits. I consider the work of significance to a breadth of fields including fire ecology, soil carbon and climate change fields. Moreover, the work provides critical experimental evidence for ecological theory more generally. Some of the conclusions are sound as presented, for example the authors demonstrated that dormancy (often considered a classic fire adaptive trait) is only applicable in a narrow range of wildfire situations. I also believe the proposed ecological framework (fig 8) is broadly correct. However, there are several avenues that this paper explores that hinge of the authors initial conceptualisation of their traits, which I believe needs revisiting (my rationale is detailed below). I also find more clarity in the presentation of some key methodology is needed before publication can be recommended.

The authors propose three microbial traits- fire survival, fast growth, and affinity for the post-fire environment which they roughly map to Malik et al's 2019 Yield-Acquisition-Stress tolerance framework. While the authors rationalizations for these mappings make sense at a glance, there are a few limitations in how traits have been assigned against the YAS system that impact their hypotheses. The main issues is that the YAS system is specifically for unpacking C cycling strategies - and dormancy (one of the authors traits) is not an C cycling strategy. The second issues is that the YAS system does not encapsulate definitions that consider temporal fluctuations in resource availability (i.e. disturbance, for example due to fire). Indeed, Malik et al highlights that one of the challenges of applying trait frameworks to microbes arises from the complexity of microbial characteristics across space and time.

Based on their allocation to the YAS framework, the authors hypothesize that trade-offs will exist between these three traits. While it is true that dormancy strategies such as spore formation are generally hypothesized to be aligned with S-selection as they fit with the Grimesian definition of 'traits

7facilitating survival...'. Malik et al considers S-selected traits to be those that divert carbon from growth to detoxification strategies. Indeed, I would argue that under Maliks and Grimes original Competitor-Stress tolerator-Ruderal (CSR) definitions (from which YAS is derived), traits are only truly S-selected if an organism is still actively growing and contributing to the wider ecosystem. As dormancy predominantly assists growth following periods of release from resource-use constraint, selection for this trait is driven by resource variability, specifically cases where variability occurs between long periods of stability. Indeed when Grime describes ruderal biology in his 1977 paper, he highlights that "a feature of the reproductive biology of many ruderals, especially the weeds of arable fields is the ability of buried seeds to survive in the soil for long periods and, if the climate is favorable, to germinate rapidly when disturbance either exposes the seeds to light or, through removal of the insulating effect of foliage and litter,..."[1] . Following this conceptual thinking, we might hypothesize that dormancy strategies and rapid growth are two parts to the same life strategy. Indeed, we find many examples where dormancy and rapid growth are both traits associated with ruderals in the plant world (e.g. desert annuals constrained by rain, and fire ephemeral species, rainforest trees constrain by shading). Excitingly the authors own observations support this for the microbial world: In 441 : "More than half of the fire survivors, including *Bacillus* sp., were also identified as being fast growers, highlighting potential positive interactions between strategies and the possibility of dormant cells surviving and germinating post-fire, and then rapidly colonizing the burned soil."

Taken together this all suggests that the conceptualization of the authors traits (and by extension some of their hypotheses and conclusions) as currently presented are imperfect. Specifically, hypothesis 1 (that fire survival as a trait would be replaced in importance by fast growth in structuring bacterial communities) becomes problematic if these two traits are part of the same life strategy as does the hypothesis regarding trait trade-offs. These conceptual issues flow through to the testing/interpretation of this trait-based fire ecology framework in the field data. While I believe this paper has potential to be a highly impactful study, I believe the original conceptualisation upon which the working hypotheses are based need to be revisited. A potential avenue to address this it to return to Grimes' original CSR framework because disturbance is a key axis upon which Grimes' framework is built.

Data & methodology: The methodologies presented are within my field of expertise and in general I believe the methodologies to be sound however my confidence in the data is incomplete due to issues with clarity:

1. I'm still not particularly clear on sampling done for the laboratory experiments. I cannot easily reconcile (even after viewing the supp data) statements like Ln 130: "N=19 sites, N=57 organic horizons, N=36 mineral horizons for all analyses unless otherwise stated" . Why is there a different n for mineral and organic horizons. I also feel there is some assumed knowledge in these definition of soil horizons that warrants explanation for a wider audience. For example, is the organic horizon the top 10cm versus the mineral being the 10-20cm? The picture is further confused for me by statements at Ln 138 "At each site, ten soil cores were collected" I'm guessing the authors are now describing collection of the field study cores, but this is not clear to me from the text.

2. A large proportion of the work hinges on ascribing the traits 'fire survival', 'fast growth', and 'affinity for the post-fire environment', to taxa. I understand that this was done using differential

abundance testing and corncob looks like a perfectly good package to carry this out. But I cannot reconcile the authors findings with what I can see in the supplemental table 11, to which I am directed in reference to trait assignments. For example, the authors state that Ln 372 “We identified 38 post-fire environment affinity OTUs”. If I go to table S11, consider only the ASVs from the ‘affinity study’, exclude groups where soil cores < 50 degrees (as described in the methods), then take only the ASVs with a positive W statistic, then (assuming I have the direction of the differential abundance contrast correct) I am left with a list of 89 ASVs assigned to the ‘affinity’ trait. Clearly this list is too many so I can go through a process of taxonomic concatenation, and I get: 10 Phyla, 17 Classes, 41 Families or 49 Genera associated with the affinity trait. Undoubtedly, I am replicating the process of trait assignment incorrectly but I aim here to illustrate that even with the supplementary data the current description does not allow researchers to easily corroborate the authors trait assignments or check for themselves taxa associated with each trait – and I believe this would be of great interest to may researchers. Clearer description of the trait assignment process is needed in the text, including whether we are considering ASVs or a higher order concatenation of taxa. It would also be valuable to simply present the list of ASVs and their respective trait assignments to facilitate exploration of taxa that were assigned more than one trait.

More minor notes on presentation.

- OTU should really be replaced with ESV/ASV throughout. Taxonomic assignment to OTUs holds a distinct meaning and is not what was done.
- Ln 36: not -> no
- Ln 113-116. The authors note that predicted/ high throughput assignment of bacterial traits is problematic and use this to justify their trait assignment approach. Clearly there is debate in the community around this matter. Given this, the authors may consider avoiding this line of discussion all together because I don’t believe it is necessary to justify your approach. In my view it is critical that empirical trait measures are developed alongside metagenome/predictive traits. Empirical measures will always be needed to validate traits we derive from the genome (for just because a gene is predicted/present doesn’t mean it is a key determinant of an organisms environmental response). In my view your methodology is a strength in its own right for this reason. This suggestion is, of course, highly subjective and I leave it to the authors to incorporate or disregard as they see fit.
- The grey dots in figure 7 are not in the legend - are they supposed to be orange?

Sincerely,
Dr J Wood

[1]Grime, J.P., Evidence for the existence of three primary strategies in plants and its relevance to ecological and evolutionary theory. *Am. Nat.*, 1977. 111: p. 1169-1194

*****END*****

Author Rebuttal to Initial comments

Reviewers' comments:

Reviewer #1 (Remarks to the Author):

Review of Johnson et al. Nat Eco Evol

R1: The work combines field and laboratory experiments to assign bacterial traits hypothesized to be relevant for carbon fluxes and for understanding the general microbial community ecology of forest ecosystems that experience fire regimes. Strengths of the work include the complementary lab/field approach as well as the measurement of soil factors to relate to the microbiological patterns observed. I appreciated the conceptual framework with new hypotheses to suggest future directions, which is provided at the end of the work.

Response: We thank the reviewer for their favourable comments. We were also particularly excited about this paired lab/field approach, and hope it becomes more mainstream. We are also pleased to hear the reviewer finds the conceptual framework and future directions of value.

R1: The three non-mutually exclusive traits predicted to have importance (fire survival, fast growth within 5 weeks of burn, fitness post-burn at 6 months) were not observed in many taxa generally, and were not found to be highly explanatory of community outcomes or carbon fluxes. The authors concede this, and then suggest that plant community assembly and dormancy may be more important. These two alternative hypotheses are not addressed by data presented in the current work. Thus, despite the interesting and novel

10experimental design, there are not strong conclusions to be drawn because the major trait hypotheses are not supported.

Response: We appreciate the reviewer's comment here. We were somewhat surprised ourselves that survival and post-fire environmental affinity were not more clearly linked to post-fire communities in the field. That said, we would argue that the strong response of lab-identified fast growers to fire in the field (a mean of 14% of the community where burn severity index >3), and their significant correlation with burn severity ($R^2=0.27$, $p<0.001$) is a striking trend, particularly so, because it disappears entirely by five years in the field. We believe this neatly illuminates our burgeoning understanding of post-fire bacterial ecology. Additionally, even though the two other traits did not encompass large fractions of the community, they were both significantly correlated with burn severity, and have furnished interesting hypotheses about which specific traits are important post-fire. Indeed, the lack of importance of survival as a trait is somewhat at odds with popular conceptions of post-fire microbial ecology, and certainly a contrast with plant-based fire ecology. Thus, we would argue that even this quantitatively moderate result is of ecological interest.

R1: Also, there are too few taxa with each trait/combinations of traits detected to make strong predictions about trade-offs or relative strengths among the traits.

Response: We agree with the reviewer on this point, and have removed our trait trade-off figures, and now consider these trait combinations only qualitatively.

R1: There are a lot of different types of data included in this work, and as a reader I found it difficult to keep track of everything, having to often move between different sections and between main and supplemental. The manuscript could be improved with a clearer focus and by omitting results that are not directly contributing to the main outcome.

Response: We appreciate this comment, and have taken several steps to address this issue. First, we expanded the experimental overview figure to make it clearer how the lab and field studies are connected (Figure 1). Second, we collected the plots illustrating field abundances of all three traits into a single figure, rather than breaking them out by trait (Figure 4). We dropped previous Figures 5A and 6A (related to lab-only data) from the main manuscript. We also dropped Figure 7 (traits trade-off) and scaled back the focus on this part of the manuscript, as recommended by reviewers. We have tried to streamline and clarify our results and discussion wherever possible. In particular, we now group results more consistently together that are related specifically to laboratory data [Sections at lines 288, 302, 332], and field-level trait data [Sections at lines 332, 362, 374, 398, 406].

R1: Is there a possibility that how the traits were defined – based on differential DNA-based taxa abundances between appropriately controlled lab experimental conditions and then determining their collective, proportional contribution to the total community with weighted copy numbers – is somehow missing the mark for capturing them in the field? There are a few reasons why this may be.

Response: Thank you for this note. As we get into the response below, we did want to first make clear that the relative abundances of traits in the field were

not weighted by predicted copy numbers. We have made sure the text is clearer on this account. That said, we did check to see what the effect of copy number likely was on these trends, and found that the trends between trait abundance and burn severity persisted but – as would be expected – the absolute fraction of the community was reduced. Of course, since copy number can only be predicted for a subset of the community, this reduction is partly due to taxa being excluded because they cannot receive a predicted copy number, due to limitations of the ribosomal copy number database. Our more detailed response follows.

R1: One reason is that there are notably different scales of the lab v. field observations. In the lab, the taxon traits defined at differential abundances at 24 hours, 5 weeks, and 6 months, but in the field there is the 1 v. 5 years post-burn. It could be that many taxa with these traits are not observed because both field time points extend beyond the lab experiments, and so the authors are looking for persistence of taxa with these traits long term without understanding if they were detected in the field at the same time scales as the lab experiments, initially. The field is long-term assembly (press) rather than short term response (pulse). The authors introduce this in the beginning of the work, but these themes are not well integrated into the experimental design. In other words, given what is known about turnover in terrestrial microbial communities, potential historical contingency, and long-term assembly: why would we expect a taxon with a trait that supports fitness at 24 hours, 5 weeks, or 6 months after fire to be similarly fit and active, due to that same trait, at 1 and 5 years after a burn? Of the three traits examined, the only one that may be relevant at 1 and 5 years is “affinity”, but this is an extrapolation beyond the lab experimental design.

Response: This is an excellent point, but, really, this was an intentional (and necessary) part of our experimental design. To explain – we were specifically

interested in testing whether survival would remain an important trait in the post-fire community over year-long timescales. It might have been the case that the advantage obtained by differential survival immediately post-fire could have persisted in the form of enrichment over longer timescales, particularly in these boreal regions, where growing seasons can be quite cool and short. Based on ecological community assembly principles, if priority effects were important, and these surviving taxa readily grew and dominated post-fire niches, excluding or hindering the colonization of future taxa, this trait could have remained important over year-long timescales. We see many examples of this in plant communities post-fire, where over the initial years or even decades post-fire, the surviving taxa out-compete the would-be colonizers that dominate the community in later successional stages. To put it simply, we couldn't know whether this would be the case for bacteria until we ran the experiments. Then, the fact that we did find a significant enrichment of some survivors is actually quite interesting. Simultaneously, the fact that they are not a large portion of the community is also of interest, and yields an interesting reflection on the “sweet spot” of conditions in which differential survival might be important.

The other two traits were assessed at 5 weeks (fast growth) and 6 months (affinity) of incubation. While these may seem like short timescales, compared to the active / growing season of the boreal forests we were studying, they actually represent relatively large fractions of active growing time for the microbes. Both of these traits were significantly correlated with burn severity, with fast growth showing particularly interesting trends (Figure 4).

Our overarching approach was to specifically isolate and test for individual traits, not to ask general questions about who's there post-fire based purely on observations – if that had been our goal, we would have just stopped at the

field data. The incubation experiments give us valuable information about bacterial traits post-fire.

We also appreciate the notes on the pulse and press framework. We found it a valuable framing for the work, and agree that we did not adequately return to it in the discussion. We have linked it back in more directly in the Discussion at lines 440 and 507.

R1: Another reason is that the weighted copy numbers may not well-capture the nuances of changes in relative contributions (for the fast growth dataset). The literature is divided on this – some researchers urge that copy number corrections are not legitimate because of database/knowledge gaps. But when applied to construct a community weighted average, overarching bias may soften, but then patterns at the taxon level (or summed taxon level across a minor subset of the community, as here) may not emerge.

Response: This is an important point, and we see that our approach here was not quite clear. To clarify, we classified fast growers by identifying taxa that were differentially abundant 5 weeks vs. 24 hours after the burns in the lab incubations. (While, of course relative abundances can be problematic on their own, we do expect, given the burn treatments, that the community size overall was likely increasing, and so we expect that taxa that were more abundant 5 weeks post fire were actually increasing in abundance, not just persisting while other community members died.) We had previously included a sub-figure that illustrated the weighted mean predicted 16S copy number for the incubation experiment, just to show that this metric behaved as would be expected (*i.e.*, it increased between 24h and 5 weeks). However, we think this was ultimately confusing (as it just related to lab communities, not field communities), and have removed the figure from the manuscript. The field

15data figure, thus, shows the relative abundance of all “fast-grower” taxa as identified by differential abundance.

However, we agree with the reviewer that the nuances and assumptions behind the weighted 16S copy number approach are interesting and currently under consideration by the scientific community. With that in mind, we thought it would be of interest to add a figure that shows how our lab-measured fast grower relative abundance in the field corresponds to the weighted mean predicted 16S copy number in the field. Encouragingly, we find that there is good correspondence between the two, which may be of strong interest to those who use the predicted metrics to infer fast growth potential. This figure is now Figure 5.

R1: The authors have a handful of taxa that exhibit each trait in the field, and they could attempt to validate their relative contributions with a few qPCRs with specifically designed primers to determine whether this may be an issue for the current study. I am not sure if it is worth it, or if it would be better to just use relative abundances instead of applying weights. The authors could compare outcomes when using relative abundances versus weighted – if the general outcomes are the same, then it doesn't matter. If they are different, then the ultimate choice must be motivated and rationalized.

Response: This is another excellent point. We have ultimately taken the approach of using relative abundances directly. However, we had whole-community 16S qPCR data on hand for the one year post-fire dataset, where spurious effects might have been most likely. We combined the qPCR data with the relative abundance sequencing data to test whether the trends and conclusions we made using the relative abundance approach held up when

using this approach. We do find that the trends persist – there is a significant positive correlation between total abundance of fast growers and burn severity in the field. Thus, we feel somewhat more confident in relying on the relative abundance data throughout this manuscript. We have noted this analysis in the manuscript for those interested [Lines 387-393]. We also performed a similar re-analysis of the fast-grower data, accounting for predicted 16S copy number (i.e., dividing relative abundances by predicted 16S copy numbers to roughly scale them to the level of individuals), which also finds the relationship remains significant [Lines 382-387].

R1: There are a lot of interesting observations in this work for the physico-chemical properties of soil after burn and given moist/dry conditions, burn severity and different soil horizons. In some cases, the authors show that these environmental properties are deterministic/explanatory of community outcomes. There is a sense of disconnect between the environmental response and biological one that distracts from the main framing about traits but at the same time draws interest, because there are statistically supported results to discuss (whereas with the traits there are fewer results of interest...). I appreciate the challenge that exists in prioritizing and integrating all these aspects into a coherently framed piece.

Response: We thank the reviewer for their appreciation of these challenges. In general, we have used the general principle of using the incubation studies to identify taxa, and then focusing on their abundances in the field. That said, as the reviewer notes, the data from the incubation studies also hold their own interest and value. We have tried to clarify and streamline the manuscript throughout, removing much of the trait trade-off content, some of the bacterial community data that were only about the incubations (previous figures 5A and 6A). We have kept some of the environmental property data, and the respiration data from the incubation, as we agree with the reviewer that they

are of interest. We also took care to reorganize the results to make it clearer which parts were specifically focused on incubation data, and which portions were focused on relating traits to field data.

R1: The work has omitted reporting of who were the taxa that were determined to have the traits, i.e. the community structure, or general discussion of the composition/phylotypes that were observed in this system. This should be provided, at least in supplemental. A simple bar chart by treatment/condition? Also, richness/number of observed taxa and any differences between treatments/conditions? These are basic information that are expected with microbiome analyses because they help to understand the ecology of the communities and interpret their patterns.

Response: Thank you for this suggestion. We have done the following: (1) We have made sure the data table with all responders and their taxonomy is clearer (SI Table 12). (2) We have added a supplemental table that summarizes the higher-level taxonomy of all responders for trait (SI Tables 13 and 14). (3) We include a modified SI figure 11 that illustrates the fraction of total reads for each incubation study/treatment that were assigned to responders for each trait. (4) We have expanded the discussion in the main text to include brief discussion of the taxonomy of some of the most abundant / influential responders [Lines 452-469; 492-498].

Specific comments

R1: The gray scale figures do not well visually distinguish the treatments of interest, especially in e.g. Figure 2A. If grayscale must be used, suggest at least having open and closed symbols included to help to distinguish the treatments.

Response: Thanks for this suggestion, we have livened up the figures with colour throughout, using consistent colours for burn treatments and for soil horizons.

R1: L91 PyOM is introduced, leading readers to think that it will be important for the study but it not returned to with data/results. pH, however, is measured. Perhaps focus the introduction only on those pieces of information that provide context for the study. Same with L 498 in discussion- this could be streamlined a bit because PyOM was not directly measured and so speculative.

Response: This is a good point. We have cut back discussion of the PyOM in the introduction and discussion. We include some mention of it, since its potential production does square with our observations of mineralization rates and constants in the CO₂ flux data, but we have taken care to reduce the speculative aspects of this, and now discuss it more in terms of general changes to soil organic matter.

R1: It is not clear how the RNA extracts were used in this experiment to assess fire survival. They were reverse transcribed and sequenced, but then what? Did the authors apply a ratio threshold of rRNA: rDNA to determine activity? Did the authors determine differential relative abundance by rRNA (cDNA) alone? From Lines 215-216, it seems that this may be the case. If so, this would be a fundamental flaw for interpretation, as sequence counts of rRNA without reference to rDNA are not interpretable as proxies for relative abundance of the organisms. This is due to differences in gene regulation, transcription, and rDNA copy numbers across different taxa, as discussed in the literature (e.g. Blazewicz 2013 ISMEJ). I checked the github repo to try to determine how the rRNA and rDNA sequences were used, and could not deduce. The methods need to be expanded to include this information.

Response: Thank you for this note. We agree it was not sufficiently clear, and have actually changed our approach based on the reviewer's suggestion. We have clarified the text in the methods [Lines 238-242] and overhauled our R code as posted in GitHub to be clearer. First, we use differential abundance of taxa in the DNA-based reads in the burned vs. unburned laboratory samples to identify taxa that were relatively enriched 24 hours post-burn. Then, we exclude all taxa with a rRNA:DNA ratio less than 1. The remaining taxa we classify as putative fire survivors. These changes resulted in fewer total survivor OTUs being identified. However, they still represent a similar portion of the total community, and the trends and associations with burn severity in the field data are not qualitatively different.

R1: Line 222 If the cores that did not reach >50 were omitted, how many "moist" cores were left? Generally, sample sizes are not apparent in the text or reported with the figure legends. It may be useful to have a table that has all the sample sizes per experiment/condition for reference. It also would be good to plot each discrete point on the box plots (or, alternatively, use violin plots with points overlay to convey each distribution quickly). I believe that this journal requires sample sizes to be reported in the figures or figure legends.

Response: Thanks for this comment. We agree that this was not clear. We have added sample sizes in text and figure legends throughout where they were missing. To clarify: All data that correspond to the laboratory incubations on their own (*i.e.*, soil properties post-fire (Figure 2)) were included in the associated plots and analyses. Only when identifying the traits did we use the burn temperature cutoff, which we used in order to ensure most of the soil would have experienced meaningful temperature effects of the burn. This cutoff excluded all moist burns and three dry burns. We have now stated this directly in the text [Lines 246-259].

R1: L223 re: trade-offs. Only taxa with more than 1 trait can be assessed for trade-offs, but this is conceptually difficult. It could be that trade-offs are complete, but more likely, because these traits are not mutually exclusive in their underlying genes/mechanisms/regulations, that trade-offs are not biologically possible. The authors mention this possibility, but a bigger question is whether it makes sense to assume/assess trade-offs for traits that are known to be not mutually exclusive?

Response: This is an excellent point, and even one that we had come around to in the discussion to some extent. It is also in agreement with many of Reviewer 3's comments. We have majorly scaled back our consideration of trait trade-offs, removing the figure, and now only discuss potential interactions between traits in the discussion [Lines 554-564].

R1: Line 300. The “good correspondence” could be statistically supported? Could the authors report the % overlap in community composition between the field and lab (and number of taxa not detected in the lab that were detected in the field).

Response: Good point. We now report the fraction of the field communities that were detected in the lab samples that we used to infer traits (“taxa detected in the lab samples representing a mean of 93% of total reads (minimum 70%, maximum 99%) in the field samples”) [Lines 346-348].

R1: Line 304 p-value

Response: Thank you – added.

R1: Figure 3. It is hard to interpret these figures. Perhaps the lab and field samples could be separated out into their own ordinations so that the differences between the treatments can be better viewed. I assume that the reason the organic v. mineral are shown separately are because these have highest explanatory value in the community patterns? If not, please motivate.

Response: We appreciate this comment. Our primary goal with this figure was to offer a visual illustration of the overlap between lab and field community composition, thus our decision to overlay the two. However, we have now changed the symbols so that the lab symbols are closed and the field symbols are open, so it is more straightforward to distinguish the two. We no longer distinguish the organic vs. mineral horizons in different panels, since this distinction isn't our key focus, and just use shape to indicate horizons. Thank you for the suggestion.

R1: L425. Who were the other fire survivors?

Response: We now include the supplemental tables 13 and 14 where the taxonomy of all responders is summarized. The supplemental table 12 has all individual responders and is also clearer than before.

R1: L 454. “Despite this, microbial communities can take longer than five years to recover post-fire...”. The authors could quantify the degree of recovery (or similar metric of resilience) for their field dataset.

Response: Thanks for this suggestion. The resilience of the field communities alone was explored in detail in Whitman et al., 2022. However, we wanted to go further in linking community shifts to our fire traits here in this study. Thus, we added a figure indicating dissimilarity between burned and unburned field communities, one and five years post-fire, vs. relative abundance of total fire-trait taxa (Figure 6) and brief related discussion [Lines 565-576].

R1: L533 Community size (number of cells/individuals) was not directly measured? DNA concentration was, and I agree that this may proxy size/biomass, but not precisely.

Response: We agree with this. We have changed the section heading to “*Post-fire C cycling rates are not likely constrained by bacterial community*” and also updated the abstract accordingly.

R1: Given the difference in microbial nucleic acid concentration able to be extracted from different conditions (fig 4A), the community was changing in its total biomass and probably the size. But, because of this, the fraction of total reads/total read abundances that are attributed to each fire-adaptive trait (e.g., Fig S10) are challenging to think about. There could be the same number of taxa in the different conditions that produce different fractions of reads if the community size has changed.... This is not a unique issue with this study but rather a common across all sequencing studies that use relativized abundances. However, because the community biomass/size is expected to change drastically given the disturbance, it is highly relevant here for interpretation. Could these charts also be provided with a y-axis standardized to the total DNA concentration or perhaps qPCR of housekeeping or even 16S genes to enable comparisons given the differences in community biomass/sizes across the treatments? Or perhaps reporting richness will also

help to understand how the number of taxa observed varies given the conditions.

Response: We agree the interactions between community composition and size are common confounding factors in microbial community analyses. We believe that – for the core goals of identifying traits in this study – it is not likely to be leading us to spurious conclusions. For the survivors, we are directly trying to assess this effect, by seeing which taxa are differentially enriched in the burned vs. unburned samples 24h post-fire. Our core assumption here is that the reason for this differential abundance is that some (many) taxa have died in the burned samples. For the fast-growers, we compare 5 weeks to 24h post-burn. Here, we would certainly expect differences in total community size, with the 5 weeks total community size expected to increase post-burn. However, because the total community size is expected to be growing over the course of this incubation, we do not expect that we would be erroneously inferring fast growth if we identify taxa that are more relatively abundant at 5 weeks. For the affinity taxa, at 6 months post-fire, we compared burned to unburned samples – similarly to the survival samples, even if the community size had not recovered to unburned levels, here, the key question is simply which taxa are more common in the burned community, which should be detectable regardless. Additionally, we seeded all samples with a 10% by mass inoculum of unburned soil, which should have further decreased any limitations from initial community in the burned samples. That said, we appreciate the reviewer’s concern about how community may have changed within the incubated samples (*e.g.*, what is now supplemental figure S6). We have added the mean \pm SD observed OTUs for each burn treatment, incubation treatment, and horizon, to help indicate how the total community has changed in these samples (Supplemental table S11).

R1: Perhaps I missed it, but are the statistics reported that show whether

24there are differences in community structure across the 24 h, 5-week, and 6-month communities in the lab? If not, is there = some discussion to add about how communities can be overall unchanged but particular populations responsive to the treatments, and attributed due to the proposed traits?

Response: This is an interesting point. We do actually see significant differences in community composition (Supplementary Figure S6, $p < 0.001$) between the three lab incubations/durations, as would be expected. In the interest of clarity, we try not to focus most of the manuscript on analyzing the incubation communities on their own. We have added the statistical results to the ordination figure of all the lab treatments in the SI (Supplemental Figure S6).

R1: I'm not sure that Figure 7 and associated analyses are conclusive for assessing "relative strength" of traits. An improved visualization may be some version of a ternary plot, with each trait serving as the angles of the triangle? But, as far as the relative strengths, this analysis could be omitted given that there is no clear conclusion to glean from it, and omitting it may serve to further focus the manuscript. Rather than strengths/trade-offs (which seem difficult to assess given the limited number of taxa that were observed to have more than 2 traits), I was curious to know which taxa belonged to each trait and trait combination, and whether anything biologically or ecologically is known about them from the literature that may support their inclusion into those assignments.

Response: We had the same original plan! But then, there were so few taxa with all three traits that all points fell on the edges of the triangle, with none in the central space, so the figure was not effective. That said, as discussed above, we have dramatically scaled back this section of the paper. We also

have made it clearer to see which taxa are multiple responders in supplemental table 12.

R1: The rich dataset presented is a bit under-analyzed as far as relating the community assembly patterns to ecological outcomes of both community and carbon stability, resistance, resilience for each treatment (and the potential contribution of the measured traits/taxa to those properties). The work is largely descriptive despite that nice contrasts/comparisons could be quantified for recovery parameters.

Response: We appreciate this comment from the reviewer – there is a lot to consider in this dataset. We have necessarily had to limit our discussion for length, and also have not gone into detail on some of the resistance/resilience questions, which are explored in our 2022 paper. However, we liked the idea of assessing the potential contribution of the measured traits to resilience. To explore this, we have added Figure 6, which shows that the dissimilarity of the field burned communities to corresponding unburned field communities is significantly positively correlated with the fraction of the community that is represented by taxa with our identified traits. We think this figure is a nice integration of the traits and field data.

Data Availability

R1: The SUB11346170 project was not found on SRA

Response: Apologies, the dataset had just been submitted at the time; this is updated, the SRA for the incubation data is under PRJNA913093.

R1: PRJNA564811 and PRJNA825513 were found and accessible on the SRA

26Response: Thank you for checking this!

R1: The link to the ESS-DIVE data repository did not work and returned an error.

Response: The data had just been submitted but had not gone public yet. It should now be accessible at <https://data.ess-dive.lbl.gov/view/doi:10.15485/1959350>.

R1: The GitHub repository is available, and it would strongly benefit from a readme file to orient users to reproduce the analysis. The input and output files for each R script are not available, and so I could not attempt to reproduce the analyses. Importantly, the OTU tables used for analyses could be provided.

Response: Thank you for these suggestions. We have added a readme file. We have also added .csv files for the taxonomy table, sample data, and OTU table for the lab incubation. (The field data are available from their original publication.)

Reviewer #2 (Remarks to the Author):

R2: The manuscript entitled “Experimentally-determined traits shape bacterial community composition one- and five-years following a wildfire” combined field and laboratory experiments to examine the ways soil bacterial communities’ structure and function change over time following fire events in boreal forests. More specifically, the authors tested microbial survival, growth,

27and affinity, here considered to represent traits of fire response strategies, in fire-affected soils. The study was well thought and designed, and the manuscript is very well written.

Response: We thank the reviewer for their favourable comments!

R2: I was thought it particularly interesting to see a study of fire impacts that integrates functional ecology and biogeochemistry, with a nice use of field and laboratory studies to test hypotheses.

Response: Thanks to the reviewer for their comments here. We were also excited about the overall approach and hope that others may draw on the general lab-field framework used here.

R2: I was particularly excited about the experimental designs to test the consequences of changes in microbial community composition to ecosystem function (here represented as C losses as CO₂ fluxes).

Response: Thank you for this comment. We were quite interested in the CO₂ results as well.

R2: I was left wondering, however, whether the title of the paper would be more impactful if it also reflected the results on functional resilience (i.e., CO₂ flux or C mineralization).

Response: We agreed that these results were very interesting, and looked for ways to integrate them into the title previously. Our concern, in the end, was that we wanted to be clear that the CO₂ flux data are from the laboratory studies, not from the field, and did not think we could make this clear in the title while also including the key message about linking lab traits to field community composition. That said, we have taken care to make sure this finding is clear in the abstract (Lines 25, 35-37).

R2: The study was conducted using soils collected from Boreal forests and we know that fire regimes differ across ecosystem types. I had the impression that, in the discussion section, the idea that the results could be more representative of Boreal forests was a bit lost. This is not, however, to discourage the authors to place some of their discussions in the context of fires in general and terrestrial ecosystems. But I thought that placing the discussion of the results in the context of boreal forests (fuels /fire frequency/etc.) prior to discussing what the results means at the ecosystem-level in general, would be more appropriate.

Response: This is a great point. We have added text in the discussion to offer more emphasis on how these findings relate to boreal ecosystems (e.g., lines 421, 443, 495, 547, 573, 610). Many of our plant fire ecology examples are drawn from boreal systems, and we also now highlight this when making these connections.

R2: Some of the findings were that fire survival was important in structuring soil bacterial communities, although likely under specific/narrow fire conditions, and that the role of (fast) growth in structuring soil microbial communities was stronger at the first-year post-fire than five years following

the fire. Also, microbial community size and composition did not alter soil CO₂ fluxes following fires. Soil conditions were considered in lines 479-486, for example, but I do think that both fire and soil conditions are driving these trends.

Response: We agree with this, and try to include consideration of both fire effects as well as post-fire soil properties in our interpretation of the microbial community response.

R2: Regarding statements in line 521- 524, I agree that is very hard to disregard the importance of these interactions belowground during ecosystem recovery trajectory.

Response: We agreed that the consideration of microbial traits and the interactions between them are important for understanding post-fire ecology belowground. We have scaled back our discussion of trait trade-offs somewhat, but do consider how traits may interact to some extent [Lines 554-576].

R2: Boreal forest soils are considered a long reservoir of carbon. Fire could affect this reservoir by affecting the soil environment where microbes regulate the cycling of carbon. But we still need to better understand under what conditions and why fires affect release of this stored carbon to the atmosphere in the future. The paper by Johnson et al. contributes to knowledge of how microbes are affected by fires, shedding light to the ecological mechanisms that are more likely to prevail and affect carbon storage during the recovery trajectory following a fire.

Response: We are glad the reviewer appreciated this paper – we agree there is lots to be learned about the specific mechanisms behind microbial C mineralization post-fire.

Reviewer #3 (Remarks to the Author):

R3: This study thoroughly sheds light on the ecology of post-fire bacterial communities through the lens of adaptive traits. I consider the work of significance to a breadth of fields including fire ecology, soil carbon and climate change fields.

Response: Thanks to the reviewer for their comments. We are also excited about what this paper might offer across fields.

R2: Moreover, the work provides critical experimental evidence for ecological theory more generally. Some of the conclusions are sound as presented, for example the authors demonstrated that dormancy (often considered a classic fire adaptive trait) is only applicable in a narrow range of wildfire situations. I also believe the proposed ecological framework (fig 8) is broadly correct.

Response: We are glad the reviewer appreciated this approach. As just a short note, while we would assume that many of the identified survivors would have been in dormant stages during the fire, we did not explicitly test for dormancy, per se.

R2: However, there are several avenues that this paper explores that hinge of the authors initial conceptualisation of their traits, which I believe needs revisiting (my rational is detailed below). I also find more clarity in the presentation of some key methodology is needed before publication can be recommended.

Response: Thanks to the reviewer for their careful consideration of the manuscript and thoughtful suggestions. We address each one below in-line.

R2: The authors propose three microbial traits– fire survival, fast growth, and affinity for the post-fire environment which they roughly map to Malik et al's 2019 Yield-Acquisition-Stress tolerance framework. While the authors rationalizations for these mappings make sense at a glance, there are a few limitations in how traits have been assigned against the YAS system that impact their hypotheses. The main issues is that the YAS system is specifically for unpacking C cycling strategies - and dormancy (one of the authors traits) is not an C cycling strategy. The second issues is that the YAS system does not encapsulate definitions that consider temporal fluctuations in resource availability (i.e. disturbance, for example due to fire). Indeed, Malik et al highlights that one of the challenges of applying trait frameworks to microbes arises from the complexity of microbial characteristics across space and time.

Based on their allocation to the YAS framework, the authors hypothesize that trade-offs will exist between these three traits. While it is true that dormancy strategies such as spore formation are generally hypothesized to be aligned with S-selection as they fit with the Grimesian definition of 'traits facilitating survival...'. Malik et al considers S-selected traits to be those that divert carbon from growth to detoxification strategies. Indeed, I would argue that under Maliks and Grimes original Competitor-Stress tolerator-Ruderal (CSR) definitions (from which YAS is derived), traits are only truly S-selected if an

32

organism is still actively growing and contributing to the wider ecosystem. As dormancy predominantly assists growth following periods of release from resource-use constraint, selection for this trait is driven by resource variability, specifically cases where variability occurs between long periods of stability. Indeed when Grime describes ruderal biology in his 1977 paper, he highlights that “a feature of the reproductive biology of many ruderals, especially the weeds of arable fields is the ability of buried seeds to survive in the soil for long periods and, if the climate is favorable, to germinate rapidly when disturbance either exposes the seeds to light or, through removal of the insulating effect of foliage and litter,...”[1] . Following this conceptual thinking, we might hypothesize that dormancy strategies and rapid growth are two parts to the same life strategy. Indeed, we find many examples where dormancy and rapid growth are both traits associated with ruderals in the plant world (e.g. desert annuals constrained by rain, and fire ephemeral species, rainforest trees constrain by shading). Excitingly the authors own observations support this for the microbial world: In 441 :“More than half of the fire survivors, including *Bacillus* sp., were also identified as being fast growers, highlighting potential positive interactions between strategies and the possibility of dormant cells surviving and germinating post-fire, and then rapidly colonizing the burned soil.”

Taken together this all suggests that the conceptualization of the authors traits (and by extension some of their hypotheses and conclusions) as currently presented are imperfect. Specifically, hypothesis 1 (that fire survival as a trait would be replaced in importance by fast growth in structuring bacterial communities) becomes problematic if these two traits are part of the same life strategy as does the hypothesis regarding trait trade-offs. These conceptual issues flow through to the testing/interpretation of this trait-based fire ecology framework in the field data. While I believe this paper has potential to be a highly impactful study, i believe the original conceptualisation upon which the working hypotheses are based need to be revisited. A potential avenue to

address this it to return to Grimes' original CSR framework because disturbance is a key axis upon which Grimes' framework is built.

Response: We offer sincere thanks to the reviewer for their thoughtful consideration and explanation of these limitations to our previous conception of the traits and their relationships. We take them to heart, and have made many adjustments, as follows.

First, we agree with the suggestion that the hypothesis that trait trade-offs should necessarily exist is somewhat problematic. In fact, at the very initial conceptualization of the study, we had been wondering as much about “super-responders” who might possess multiple fire-related traits. We have been happy to remove the framing of traits trade-offs as something that is explicitly expected and even tested for (*i.e.*, removing the intro text about trade-offs, removing our hypothesis about trade-offs, removing the corresponding figure, and much of the discussion). With the dropping of the trade-off hypotheses, we have also changed hypothesis 1 to explicitly recognize that the traits may not be mutually exclusive [Lines 112-120].

We also agree that the taxa with multiple traits are of clear interest. In general, we have tried to take a conservative approach to identifying the traits, taking only strong responders under clearly burned conditions. Thus, it is perhaps not surprising that we did not identify many taxa that had multiple traits. In further refining our approach in response to reviewers, we ended up with fewer fast growers than we had initially, so now only 10% of our survivors are also identified as fast-growers. We discuss these taxa and potential relationships between traits at lines 463-469 and lines 554-564.

Finally, we appreciate the distinctions behind the YAS framework and its specific limitations to C use. Given these insights, we have updated our introduction to focus more directly on the original Grime's framework, explicitly discussing how each of our strategies relates to those of Grime [Lines 66-67; 83-87; 94-98; 107-111], and relating our discussion back to these concepts as well [Lines 463-467 and 494-496].

R2: Data & methodology: The methodologies presented are within my field of expertise and in general I believe the methodologies to be sound however my confidence in the data is incomplete due to issues with clarity:

1. I'm still not particularly clear on sampling done for the laboratory experiments. I cannot easily reconcile (even after viewing the supp data) statements like Ln 130: "N=19 sites, N=57 organic horizons, N=36 mineral horizons for all analyses unless otherwise stated". Why is there a different n for mineral and organic horizons. I also feel there is some assumed knowledge in these definition of soil horizons that warrants explanation for a wider audience. For example, is the organic horizon the top 10cm versus the mineral being the 10-20cm? The picture is further confused for me by statements at Ln 138 "At each site, ten soil cores were collected" I'm guessing the authors are now describing collection of the field study cores, but this is not clear to me from the text.

Response: Thank you for these notes. We have clarified the experimental design (Lines 150-155; Figure 1) and made the N for each figure explicit in the figure captions. Specifically, for the soils, yes, this is a great point that non boreal soil scientists might need clarification on. Briefly, we chose sites to represent a broad swathe of vegetation and soils typical of this region. Some soils – largely those on which black spruce is dominant have very thick organic horizons (or may even be entirely organic soils). Thus, our sampling

35of any of these sites only includes organic soil horizons. We sampled by horizon rather than just by depth, because there are large discontinuities in soil properties across a horizon. If we sampled by depth, we would be ignoring these important differences, mixing horizons together. A very clear example of this is in supplemental figure S2, where you can see that the organic horizons have very high C contents compared to the mineral horizons.

To clarify the ten cores – we collected cores from the field for the laboratory studies. Because there can be high spatial variability, and for insurance in general, we collected ten cores at each site. However, we only used three cores (dry burn, moist burn, control) for any given site. We chose the cores with the most similar horizon thicknesses for our three treatments for each site. However, we recognize this was confusing, and now just mention the three cores that we ultimately used in our methods.

R2: 2. A large proportion of the work hinges on ascribing the traits ‘fire survival’, ‘fast growth’, and ‘affinity for the post-fire environment’, to taxa. I understand that this was done using differential abundance testing and corncob looks like a perfectly good package to carry this out. But I cannot reconcile the authors findings with what I can see in the supplemental table 11, to which I am directed in reference to trait assignments. For example, the authors state that Ln 372 “We identified 38 post-fire environment affinity OTUs”. If I go to table S11, consider only the ASVs from the ‘affinity study’, exclude groups where soil cores < 50 degrees (as described in the methods), then take only the ASVs with a positive W statistic, then (assuming I have the direction of the differential abundance contrast correct) I am left with a list of 89 ASVs assigned to the ‘affinity’ trait. Clearly this list is too many so I can go through a process of taxonomic concatenation, and I get: 10 Phyla, 17 Classes, 41 Families or 49 Genera associated with the affinity trait. Undoubtedly, I am replicating the process of trait assignment incorrectly but I

aim here to illustrate that even with the supplementary data the current description does not allow researchers to easily corroborate the authors trait assignments or check for themselves taxa associated with each trait – and I believe this would be of great interest to many researchers. Clearer description of the trait assignment process is needed in the text, including whether we are considering ASVs or a higher order concatenation of taxa. It would also be valuable to simply present the list of ASVs and their respective trait assignments to facilitate exploration of taxa that were assigned more than one trait.

Response: Thank you for these suggestions, and our apologies for the initial lack of clarity. Briefly, initially, we included both strong and weak responders in the SI list, but only used strong responders in the manuscript analyses (partly to quantify the potential trait trade-offs, which we have now largely cut). We recognize that was confusing, and have clarified and streamlined our approach to trait identification in the methods. In the process, our final taxa numbers have changed, but the broad trends remain very similar (*i.e.*, the taxa that were dropped represented only small fractions of the total communities). The exact details are now more clearly explained in the methods [Lines 236-255]. Moreover, the supplemental table 12 is now clearer and includes the exact taxa considered throughout the manuscript.

R2: More minor notes on presentation.

R2: - OTU should really be replaced with ESV/ASV throughout. Taxonomic assignment to OTUs holds a distinct meaning and is not what was done.

Response: We appreciate this perspective, but argue that OTU is a better term in this context, for a few reasons. First, the term OTU is broad. Dating

back to at least Sneath and Sokol (1962), the term is used to refer simply to taxonomic entities identified based on some phenetic resemblance, and has been employed in many microbiological contexts beyond amplicon sequencing (e.g., T-RFLP or DGGE). In our opinion, there is no need to shift to a new term specifically for this narrow definition of an amplicon-sequencing-based OTU designated using the dada2 algorithm or another algorithm that includes little to no clustering. All types of OTUs are operationally defined, as are ASVs. Furthermore, we would argue that it's useful to remind ourselves of the limitations (operational definition) of ASVs, by retaining the term OTU. Two important examples include: (1) taxa with identical 16S sequences over the sequenced region (i.e., identical ASVs) could still be ecologically distinct; (2) taxa with multiple copies of the rRNA gene, that have single nucleotide differences in this gene, can result in a single individual being classified as two separate ASVs (Schloss, 2021, mSphere, <https://doi.org/10.1128/mSphere.00191-21>). Thus, unless the style guide of *Nature Ecology and Evolution* requires otherwise, we would respectfully prefer to use the term OTU. We do also give a nod to the term ASV at line 282.

R2: - Ln 36: not -> no

Response: Thank you, this sentence is now deleted because it referred to the trait trade-offs.

R2: - Ln 113-116. The authors note that predicted/ high throughput assignment of bacterial traits is problematic and use this to justify their trait assignment approach. Clearly there is debate in the community around this matter. Given this, the authors may consider avoiding this line of discussion all together because I don't believe it is necessary to justify your approach. In my view it is critical that empirical trait measures are developed alongside

38metagenome/predictive traits. Empirical measures will always be needed to validate traits we derive from the genome (for just because a gene is predicted/present doesn't mean it is a key determinant of an organisms environmental response). In my view your methodology is a strength in its own right for this reason. This suggestion is, of course, highly subjective and I leave it to the authors to incorporate or disregard as they see fit.

Response: We appreciate the reviewer's thoughts on this, and have moderated our tone in this sentence. It now reads, "While ecological predictions can be made based on the genetic features of a given organism or community, the extraordinary diversity of soil bacteria [40], the vast majority of which remain uncultured [41], and phenomena such as horizontal gene transfer impair our ability to use taxonomy alone to confidently infer bacterial traits [42]. Thus, we used an uncommon approach to assign traits to bacteria in a high-throughput manner in this study (Fig. 1) [...]"

R2: - The grey dots in figure 7 are not in the legend - are they supposed to be orange?

Response: Our apologies for this error. The figure is now gone (they indicated taxa that were weak but statistically significant negative responders for a given trait).

Sincerely,
Dr J Wood

[1]Grime, J.P., Evidence for the existence of three primary strategies in plants and its relevance to ecological and evolutionary theory. *Am. Nat.*, 1977. 111:

39p. 1169-1194

Decision Letter, first revision:

6th April 2023

Dear Dr. Whitman,

Thank you for submitting your revised manuscript "Experimentally-determined traits shape bacterial community composition one and five years following wildfire" (NATECOLEVOL-221218147A). It has now been seen again by the original reviewers and their comments are below. The reviewers find that the paper has improved in revision, and therefore we'll be happy in principle to publish it in Nature Ecology & Evolution, pending minor revisions to satisfy the reviewers' final requests and to comply with our editorial and formatting guidelines.

We are now performing detailed checks on your paper and will send you a checklist detailing our editorial and formatting requirements in about two weeks. Please do not upload the final materials and make any revisions until you receive this additional information from us.

[REDACTED]

Reviewer #2 (Remarks to the Author):

I have no comments further as I believe the paper is in a good position to be published.

Reviewer #3 (Remarks to the Author):

The authors have taken care to address my key concerns. I think the removal of trait-trade-off figure and associated discussion has resulted in a more focused manuscript, also the supplemental data is clearer. I would still argue that 'fire survival' is akin to a Grimesian ruderal trait. I can see the authors have laid out their rationalisation and I acknowledge that many other microbial-CSR frameworks have drawn a similar connection between dormancy and stress tolerance. However, there remain some inconsistencies that need to be challenged to ensure the framework fit is sound and not superficial:

40Ln 97-98. The way the authors relate pulse and press disturbance to their trait assignments in this text seems to me to be at odds with definitions of pulse, press, disturbance, and ruderal. Disturbance is defined (by Grime and others) as the partial or complete destruction of biological material. So, the pulse is a disturbance, yes. And disturbance is predicted to favour ruderals/fast growers. Instead of this though, the authors choose to relate the destruction of the community to being an action that favours fire survivors. Well, I expect this is also true. Which brings me back to – survivorship could also a ruderal trait.

Further, the explanation of the press disturbance as being that which selects for fast growers does not fit with the authors original explanation of what the press disturbance is, which seems to be environmental - "changes to vegetation communities and soil properties, such as pH and C availability" -Ln 75/76. If we take this definition, then the press disturbance is not a disturbance in the Grimesian sense but instead is related to a persistent change in the physico-chemical environment. If this is so, then I would argue that it should select for key competitors - your fire thriving OTUs. As you point out elsewhere in the paper, it seems that the reason fire selects for fast growers is because it creates space (through physical destruction of other microbes) for them to grow into (i.e. due to the pulse). Not that the fire has created an optimal physical environment that favours fast growers. All of this comes back to how important it is for the three traits to represent three contrasting ecologies (this is what is implied when relating the traits to CSR/YAS but i think the authors concede that they may in fact not independent). So, I will leave the authors with this question: is it critical to the findings and conclusions of your paper that your 3 traits be fit against the three different life-strategies? The main point of Grimes (and Maliks) 3 strategies is that there is evidence they trade off against each other. We don't quite have that here and for my thinking I would have hypothesised that fire survival and rapid growth would have co-occurred. We know multiple traits contribute to a given life strategy and we know intermediate life strategies exist too, so do these three fire traits really represent 3 distinct life strategies and -more importantly - is it an issue if they don't? Why not a dichotomy of best primary colonisers versus best long term competitors? This is a question which I leave to the authors to work through.

Semantics aside, the authors have prepared an interesting and thought provoking manuscript.

Other comments

Ln 91-93. I am confused here, the text seems contradictory as it first states that:

"a decrease in 16S ribosomal RNA (rRNA) copy number from 4 months to 2.5 years post-burn ... correlates with faster growth rates [29, 30]."

And then goes on to say:

"Fast-growing bacteria with higher rRNA gene copy numbers may have a selective advantage in responding to increased nutrient availability post-fire [31]."

Did the author mean to say increase in copy number in the first instance?

41Figure 2 – is there a unit that can be added for pool size and decay rate?

Figure 6 - I think this is has not strictly been referenced in the results text.

Both the results and discussion allude to a significant relationship between burn severity and fire survivorship, but there is no p-value presented on figure 4A for 5-year data?

Our ref: NATECOLEVOL-221218147A

4th May 2023

Dear Dr. Whitman,

Thank you for your patience as we've prepared the guidelines for final submission of your Nature Ecology & Evolution manuscript, "Experimentally-determined traits shape bacterial community composition one and five years following wildfire" (NATECOLEVOL-221218147A). Please carefully follow the step-by-step instructions provided in the attached file, and add a response in each row of the table to indicate the changes that you have made. Please also check and comment on any additional marked-up edits we have proposed within the text. Ensuring that each point is addressed will help to ensure that your revised manuscript can be swiftly handed over to our production team.

****We would like to start working on your revised paper, with all of the requested files and forms, as soon as possible (preferably within two weeks). Please get in contact with us immediately if you anticipate it taking more than two weeks to submit these revised files.****

42In recognition of the time and expertise our reviewers provide to Nature Ecology & Evolution's editorial process, we would like to formally acknowledge their contribution to the external peer review of your manuscript entitled "Experimentally-determined traits shape bacterial community composition one and five years following wildfire". For those reviewers who give their assent, we will be publishing their names alongside the published article.

Nature Ecology & Evolution offers a Transparent Peer Review option for new original research manuscripts submitted after December 1st, 2019. As part of this initiative, we encourage our authors to support increased transparency into the peer review process by agreeing to have the reviewer comments, author rebuttal letters, and editorial decision letters published as a Supplementary item. When you submit your final files please clearly state in your cover letter whether or not you would like to participate in this initiative. Please note that failure to state your preference will result in delays in accepting your manuscript for publication.

Cover suggestions

As you prepare your final files we encourage you to consider whether you have any images or illustrations that may be appropriate for use on the cover of Nature Ecology & Evolution.

Nature Ecology & Evolution has now transitioned to a unified Rights Collection system which will allow our Author Services team to quickly and easily collect the rights and permissions required to publish your work. Approximately 10 days after your paper is formally accepted, you will receive an email in providing you with a link to complete the grant of rights. If your paper is eligible for Open Access, our Author Services team will also be in touch regarding any additional information that may be required to arrange payment for your article.

Please note that Nature Ecology & Evolution is a Transformative Journal (TJ). Authors may publish their research with us through the traditional subscription access route or make their paper immediately open access through payment of an article-processing charge (APC). Authors will not be required to make a final decision about access to their article until it has been accepted. Find out more

43about Transformative Journals

Authors may need to take specific actions to achieve [compliance with funder and institutional open access mandates](https://www.springernature.com/gp/open-research/funding/policy-compliance-faqs). If your research is supported by a funder that requires immediate open access (e.g. according to [Plan S principles](https://www.springernature.com/gp/open-research/plan-s-compliance)) then you should select the gold OA route, and we will direct you to the compliant route where possible. For authors selecting the subscription publication route, the journal's standard licensing terms will need to be accepted, including [self-archiving and license to publish](https://www.nature.com/nature-portfolio/editorial-policies/self-archiving-and-license-to-publish). Those licensing terms will supersede any other terms that the author or any third party may assert apply to any version of the manuscript.

[REDACTED]

[REDACTED]

Reviewer #2:

Remarks to the Author:

I have no comments further as I believe the paper is in a good position to be published.

Reviewer #3:

Remarks to the Author:

The authors have taken care to address my key concerns. I think the removal of trait-trade-off figure and associated discussion has resulted in a more focused manuscript, also the supplemental data is clearer. I would still argue that 'fire survival' is akin to a Grimesian ruderal trait. I can see the authors have laid out their rationalisation and I acknowledge that many other microbial-CSR frameworks have drawn a similar connection between dormancy and stress tolerance. However, there remain some

44inconsistencies that need to be challenged to ensure the framework fit is sound and not superficial:

Ln 97-98. The way the authors relate pulse and press disturbance to their trait assignments in this text seems to me to be at odds with definitions of pulse, press, disturbance, and ruderal. Disturbance is defined (by Grime and others) as the partial or complete destruction of biological material. So, the pulse is a disturbance, yes. And disturbance is predicted to favour ruderals/fast growers. Instead of this though, the authors choose to relate the destruction of the community to being an action that favours fire survivors. Well, I expect this is also true. Which brings me back to – survivorship could also a ruderal trait.

Further, the explanation of the press disturbance as being that which selects for fast growers does not fit with the authors original explanation of what the press disturbance is, which seems to be environmental - "changes to vegetation communities and soil properties, such as pH and C availability" -In 75/76. If we take this definition, then the press disturbance is not a disturbance in the Grimesian sense but instead is related to a persistent change in the physico-chemical environment. If this is so, then I would argue that it should select for key competitors - your fire thriving OTUs. As you point out elsewhere in the paper, it seems that the reason fire selects for fast growers is because it creates space (through physical destruction of other microbes) for them to grow into (i.e. due to the pulse). Not that the fire has created an optimal physical environment that favours fast growers. All of this comes back to how important it is for the three traits to represent three contrasting ecologies (this is what is implied when relating the traits to CSR/YAS but i think the authors concede that they may in fact not independent). So, I will leave the authors with this question: is it critical to the findings and conclusions of your paper that your 3 traits be fit against the three different life-strategies? The main point of Grimes (and Maliks) 3 strategies is that there is evidence they trade off against each other. We don't quite have that here and for my thinking I would have hypothesised that fire survival and rapid growth would have co-occurred. We know multiple traits contribute to a given life strategy and we know intermediate life strategies exist too, so do these three fire traits really represent 3 distinct life strategies and -more importantly - is it an issue if they don't? Why not a dichotomy of best primary colonisers versus best long term competitors? This is a question which I leave to the authors to work through.

Semantics aside, the authors have prepared an interesting and thought provoking manuscript.

Other comments

Ln 91-93. I am confused here, the text seems contradictory as it first states that:

"a decrease in 16S ribosomal RNA (rRNA) copy number from 4 months to 2.5 years post-burn ... correlates with faster growth rates [29, 30]."

And then goes on to say:

"Fast-growing bacteria with higher rRNA gene copy numbers may have a selective advantage in responding to increased nutrient availability post-fire [31]."

45Did the author mean to say increase in copy number in the first instance?

Figure 2 – is there a unit that can be added for pool size and decay rate?

Figure 6 - I think this is has not strictly been referenced in the results text.

Both the results and discussion allude to a significant relationship between burn severity and fire survivorship, but there is no p-value presented on figure 4A for 5-year data?

Author Rebuttal, first revision:

Response to Reviewers

Reviewer #2:

Remarks to the Author:

I have no comments further as I believe the paper is in a good position to be published.

Response: We thank Reviewer 2 for their helpful suggestions throughout this process.

Reviewer #3:

Remarks to the Author:

R3: The authors have taken care to address my key concerns. I think the removal of trait-trade-off figure and associated discussion has resulted in a more focused manuscript, also the supplemental data is clearer.

Response: We are glad the reviewer finds our revisions appropriate and agrees that they improve the manuscript.

R3: I would still argue that 'fire survival' is akin to a Grimesian ruderal trait. I can see the authors have laid out their rationalisation and I acknowledge that many other microbial-CSR frameworks have drawn a similar connection between dormancy and stress tolerance.

Response: We appreciate the reviewer's point and also their flexibility.

R3: However, there remain some inconsistencies that need to be challenged to ensure the framework fit is sound and not superficial:
Ln 97-98. The way the authors relate pulse and press disturbance to their trait assignments in this text seems to me to be at odds with definitions of pulse, press, disturbance, and ruderal. Disturbance is defined (by Grime and others) as the partial or complete destruction of biological material.

Response: This is an important point. While we draw substantially on Grime's framework, the definition of disturbance we are using for the purposes of this paper is not equivalent to Grime's definition of disturbance. Rather, it is *sensu* Shade et al. [12] – *i.e.*, “causal events that either (1) alter the immediate environment and have possible repercussions for a community or (2) directly alter a community [13, 14].” Since there are various definitions of disturbance in the field of ecology, we have amended the manuscript to ensure we directly define the term for the purposes of this paper [Lines 62-64].

R3: So, the pulse is a disturbance, yes. And disturbance is predicted to favour ruderals/fast growers. Instead of this though, the authors choose to relate the destruction of the community to being an action that favours fire survivors.

Well, I expect this is also true. Which brings me back to – survivorship could also a ruderal trait.

Further, the explanation of the press disturbance as being that which selects for fast growers does not fit with the authors original explanation of what the press disturbance is, which seems to be environmental - “changes to vegetation communities and soil properties, such as pH and C availability” -In 75/76. If we take this definition, then the press disturbance is not a disturbance in the Grimesian sense but instead is related to a persistent change in the physico-chemical environment. If this is so, then I would argue that it should select for key competitors - your fire thriving OTUs. As you point out elsewhere in the paper, it seems that the reason fire selects for fast growers is because it creates space (through physical destruction of other microbes) for them to grow into (i.e. due to the pulse). Not that the fire has created an optimal physical environment that favours fast growers.

Response: We would agree with the reviewer’s characterizations in the context of Grime’s definition of disturbance, and have taken care now to directly define disturbance more broadly for the purposes of this paper. Additionally, we appreciate the notes on our representation of pulse and press aspects of fire disturbance and how they would relate to our three traits of interest. We have edited these sections to make it clearer that we are not arguing that press disturbances alone are driving the fast growth response in particular [Lines 73-94].

R3: All of this comes back to how important it is for the three traits to represent three contrasting ecologies (this is what is implied when relating the traits to CSR/YAS but i think the authors concede that they may in fact not independent). So, I will leave the authors with this question: is it critical to the

findings and conclusions of your paper that your 3 traits be fit against the three different life-strategies? The main point of Grimes (and Maliks) 3 strategies is that there is evidence they trade off against each other. We don't quite have that here and for my thinking I would have hypothesised that fire survival and rapid growth would have co-occurred. We know multiple traits contribute to a given life strategy and we know intermediate life strategies exist too, so do these three fire traits really represent 3 distinct life strategies and -more importantly - is it an issue if they don't? Why not a dichotomy of best primary colonisers versus best long term competitors? This is a question which I leave to the authors to work through.

Semantics aside, the authors have prepared an interesting and thought provoking manuscript.

Response: These are also excellent points. In the end, we are most interested in identifying fire-related traits, while also relating them to and recognizing the wide body of work in ecology on disturbance, traits, and trade-offs. We now return to this idea in the discussion more directly, explicitly noting the differences between our chosen traits and Grime's, and that trade-offs may not be expected [Lines 381-390].

Other comments

R3: Ln 91-93. I am confused here, the text seems contradictory as it first states that:

“a decrease in 16S ribosomal RNA (rRNA) copy number from 4 months to 2.5 years post-burn ... correlates with faster growth rates [29, 30].”

And then goes on to say:

49“Fast-growing bacteria with higher rRNA gene copy numbers may have a selective advantage in responding to increased nutrient availability post-fire [31].”

Did the author mean to say increase in copy number in the first instance?

Response: We agree this sentence was worded confusingly. We have revised both sentences as follows: “This is supported by previous work – higher 16S ribosomal RNA (rRNA) gene copy numbers are correlated with faster growth rates, and higher 16S ribosomal RNA (rRNA) gene copy numbers were observed at 4 months than at 2.5 years post-burn. Fast-growing bacteria with higher rRNA gene copy numbers may have a selective advantage in responding to increased nutrient availability post-fire.”

R3: Figure 2 – is there a unit that can be added for pool size and decay rate?

Response: Good point – the decay rate should have units of days⁻¹. We have amended this. Thank you! The pool size is relative (between 0 and 1) and does not have units.

R3: Figure 6 - I think this is has not strictly been referenced in the results text.

Response: Thank you for catching this. We have added the in-text reference.

R3: Both the results and discussion allude to a significant relationship between burn severity and fire survivorship, but there is no p-value presented on figure 4A for 5-year data?

Response: We had only included it once, since the models include 1 and 5 year data (and the interaction term, where relevant), but recognize this was confusing as presented in the figure. We now report the P and R^2 for the 5-year data in the panels.

Final Decision Letter:

26th June 2023

Dear Dr Whitman,

We are pleased to inform you that your Article entitled "Experimentally-determined traits shape bacterial community composition one and five years following wildfire", has now been accepted for publication in Nature Ecology & Evolution.

Over the next few weeks, your paper will be copyedited to ensure that it conforms to Nature Ecology and Evolution style. Once your paper is typeset, you will receive an email with a link to choose the appropriate publishing options for your paper and our Author Services team will be in touch regarding any additional information that may be required

You will not receive your proofs until the publishing agreement has been received through our system

Due to the importance of these deadlines, we ask you please us know now whether you will be difficult to contact over the next month. If this is the case, we ask you provide us with the contact information (email, phone and fax) of someone who will be able to check the proofs on your behalf, and who will be available to address any last-minute problems . Once your paper has been scheduled for online publication, the Nature press office will be in touch to confirm the details.

Acceptance of your manuscript is conditional on all authors' agreement with our publication policies (see www.nature.com/authors/policies/index.html). In particular your manuscript must not be published elsewhere and there must be no announcement of the work to any media outlet until the publication date (the day on which it is uploaded onto our web site).

Please note that *Nature Ecology & Evolution* is a Transformative Journal (TJ). Authors may publish their research with us through the traditional subscription access route or make their paper

51immediately open access through payment of an article-processing charge (APC). Authors will not be required to make a final decision about access to their article until it has been accepted. [Find out more about Transformative Journals](https://www.springernature.com/gp/open-research/transformative-journals)

Authors may need to take specific actions to achieve [compliance with funder and institutional open access mandates](https://www.springernature.com/gp/open-research/funding/policy-compliance-faqs). If your research is supported by a funder that requires immediate open access (e.g. according to [Plan S principles](https://www.springernature.com/gp/open-research/plan-s-compliance)) then you should select the gold OA route, and we will direct you to the compliant route where possible. For authors selecting the subscription publication route, the journal's standard licensing terms will need to be accepted, including [self-archiving-and-license-to-publish](https://www.nature.com/nature-portfolio/editorial-policies/self-archiving-and-license-to-publish). Those licensing terms will supersede any other terms that the author or any third party may assert apply to any version of the manuscript.

We welcome the submission of potential cover material (including a short caption of around 40 words) related to your manuscript; suggestions should be sent to Nature Ecology & Evolution as electronic files (the image should be 300 dpi at 210 x 297 mm in either TIFF or JPEG format). Please note that such pictures should be selected more for their aesthetic appeal than for their scientific content, and that colour images work better than black and white or grayscale images. Please do not try to design a cover with the Nature Ecology & Evolution logo etc., and please do not submit composites of images related to your work. I am sure you will understand that we cannot make any promise as to whether any of your suggestions might be selected for the cover of the journal.

To assist our authors in disseminating their research to the broader community, our SharedIt initiative

52provides you with a unique shareable link that will allow anyone (with or without a subscription) to read the published article. Recipients of the link with a subscription will also be able to download and print the PDF.

You can generate the link yourself when you receive your article DOI by entering it here: http://authors.springernature.com/share.

[REDACTED]

P.S. Click on the following link if you would like to recommend Nature Ecology & Evolution to your librarian <http://www.nature.com/subscriptions/recommend.html#forms>

** Visit the Springer Nature Editorial and Publishing website at www.springernature.com/editorial-and-publishing-jobs for more information about our career opportunities. If you have any questions please click here. **